**Submission**

# Numerical simulations of inflationary dynamics: slow roll and beyond

**Siddharth S. Bhatt**[1*], **Swagat S. Mishra**[2#] (iD), **Soumen Basak**[1†], **and Surya N. Sahoo**[3‡]

**1** Indian Institute of Science Education and Research, Thiruvananthapuram 695551, India
**2** School of Physics and Astronomy, University of Nottingham, Nottingham, NG7 2RD, UK.
**3** National Institute of Science Education and Research, Bhubaneswar, Odisha 752050, India.

* bhattsiddharth17@alumni.iisertvm.ac.in , # swagat.mishra@nottingham.ac.uk ,
† sbasak@iisertvm.ac.in , ‡ suryans@niser.ac.in

## Abstract

Numerical simulations of the inflationary dynamics are presented here for a single canonical scalar field minimally coupled to gravity. We spell out the basic equations governing the inflationary dynamics in terms of cosmic time $t$ and define a set of dimensionless variables convenient for numerical analysis. We then provide a link to our simple numerical `Python` code on `GitHub` that can be used to simulate the background dynamics as well as the evolution of linear perturbations during inflation. The code computes both scalar and tensor power spectra for a given inflaton potential $V(\phi)$. We discuss a concrete algorithm to use the code for various purposes, especially for computing the enhanced scalar power spectrum in the context of Primordial Black Holes and scalar-induced Gravitational Waves. We also compare the efficiency of different variables used in the literature to compute the scalar fluctuations. We intend to extend the framework to simulate the dynamics of a number of different quantities, including the computation of scalar-induced second-order tensor power spectrum in the near future.

# 1   Introduction

Cosmic inflation has emerged as the leading scenario for describing the very early universe prior to the commencement of the radiative hot Big Bang Phase [1–6]. According to the inflationary paradigm, a transient epoch of at least 60-70 e-folds of rapid accelerated expansion suffices in setting natural initial conditions for the background space-time in the form of spatial flatness as well as statistical homogeneity and isotropy on large angular scales [2–4, 7]. Additionally, (and more significantly,) quantum fluctuations during inflation naturally generate a spectrum of almost scale-invariant initial scalar fluctuations which seed the temperature and polarisation fluctuations in the Cosmic Microwave Background (CMB) Radiation, and later, the formation of structure in the universe [7–11]. In addition to scalar perturbations, quantum fluctuations during inflation also create a spectrum of almost scale-invariant tensor perturbations which later become gravitational waves [12, 13].

The simplest models of inflation comprising of a single scalar field, called the 'inflaton', which is minimally coupled to gravity, makes several distinct predictions [14] (*i.e* an almost scale-invariant, nearly Gaussian, and adiabatic spectrum of scalar fluctuations) most of which have received spectacular observational confirmation, particularly from the latest CMB missions [15].

However, as mentioned earlier, inflation also generates tensor perturbations that later constitute the relic gravitational wave background (GW) which imprints a distinct signature on the CMB power spectrum in the form of the B-mode polarization [15]. The amplitude of these relic GWs provides us information about the inflationary energy scale while their spectrum enables us to access general properties of the epoch of reheating, being exceedingly sensitive to the post-inflationary equation of state [13, 16]. The amplitude of inflationary tensor fluctuations, relative to that of scalar fluctuations, is usually characterised by the tensor-to-scalar ratio $r$. Different models of inflation predict different values for $r$ which is sensitive to the gradient of the inflaton potential $V_{,\phi}(\phi) = \frac{dV(\phi)}{d\phi}$ relative to its height $V(\phi)$. Convex potentials predict large values for $r$, while concave potentials predict relatively small values of $r$. While the spectrum of inflationary tensor fluctuations has not yet been observed, current CMB observations are able to place an upper bound on the tensor-to-scalar ratio on large angular scales. In particular, the latest CMB observations of BICEP/Keck [17], combined with those of the PLANCK mission [15], place the strong upper bound $r \leq 0.036$ (at 95% confidence).

This most recent upper bound on $r$ has important consequences for single field canonical inflation. In particular, given $r \leq 0.036$, all monotonically increasing convex potentials, including the whole family of monomial potentials $V(\phi) \propto \phi^p$, are completely ruled out in the canonical framework. Among these strongly disfavoured models are the simplest classic inflaton potentials $\frac{1}{2}m^2\phi^2$ and $\lambda\phi^4$. Instead, the observational upper bound on $r$ appears to favour asymptotically-flat potentials possessing one or two plateau-like wings; see Refs. [18, 19]. Current observational data lead to a scenario in which the inflaton $\phi$ slowly rolls down a shallow potential $V(\phi)$ thereby giving rise to a quasi-de Sitter early stage of near-exponential expansion. A thorough analysis of the inflationary phase-space dynamics $\{\phi, \dot{\phi}\}$ for plateau potentials shows [20] that a large range of initial conditions leads to adequate inflation in these models.

However, it is important to stress that the CMB window constitutes only a tiny part of the observationally available field space between the Hubble-exit of the largest scales in the sky and the smallest scale at the end of inflation. Consequently, a substantial period of the inflationary dynamics corresponding to potentially interesting small-scale primordial physics (which accounts roughly to the last 40–50 e-folds of accelerated expansion during inflation) remains observationally unexplored, being inaccessible to the CMB and LSS observations. Any departure from the slow-roll regime, that might be triggered by a change in the dynamics of the inflaton field, would lead to interesting observational consequences on small-scales. In particular, the presence of a feature at intermediate field values that might lead to large enough amplification of the

small-scale scalar fluctuations, would facilitate the formation of Primordial Black Holes upon the Hubble re-entry of these modes during the post-inflationary epochs.

Primordial Black Holes (PBHs) are extremely interesting compact objects which might have been formed from the collapse of large density fluctuations in the early universe [21–24] and they constitute a potential candidate for dark matter [25–30]. Seeds for such large fluctuations can be generated during inflation, as mentioned above. For instance, a feature in the inflaton potential in the form of a flat inflection point can further slow down the already slowly rolling inflaton field substantially, leading to an enhancement of the primordial scalar power $P_\zeta$. A number of different features for enhancing small-scale power during inflation have been proposed in the recent years [31–49]. Hence PBHs (and the associated induced relic GWs) are excellent probes of the small-scale primordial physics.

In this paper, we discuss a simple code developed for numerical simulations of the inflationary dynamics. We introduce the relevant dimensionless variables used in our numerical analysis and provide a link to the code in our `GitHub` account, along with an `IPython notebook` as a guide for using the same. We also discuss how to use the code in various scenarios, which include phase-space analysis of inflationary initial conditions, inflationary background dynamics and determining scalar and tensor power spectra both under slow-roll approximation and beyond. The latter case has important implications for PBH formation and we discuss how to use the code to simulate the Mukhanov-Sasaki equation mode by mode. We also discuss a number of important future directions that are to be included in the forthcoming version of our paper. The primary version of our numerical framework is quite simple and less compact. It is intended to provide a pedagogical guideline for researchers who are relatively new to numerical simulations of inflation. In the forthcoming version of our work, we will introduce a much more compact numerical framework that we are currently working on which will incorporate additional new features.

This paper is organised as follows: we begin with a brief introduction of the inflationary scalar field dynamics in section 2 and quantum fluctuations in section 3. We then proceed to discuss numerical simulations of the background dynamics in section 4. This also includes studying the scalar and tensor fluctuations under the slow-roll approximations. Section 5 is dedicated to studying the inflationary quantum fluctuations by numerically solving the Mukhanov-Sasaki equation, and its application to inflaton potentials possessing a slow-roll violating feature. We carry out a comparison of the efficiency of different dimensionless variablesin our numerical analysis in Sec. 6. We also mention a number of future extensions of our numerical set-up in section 7, before concluding with a discussion section.

We work in the units $c, \hbar = 1$. The reduced Planck mass is defined to be $m_p \equiv 1/\sqrt{8\pi G} = 2.43 \times 10^{18}$ GeV. We assume the background universe to be described by a spatially flat Friedmann-Lemaitre-Robertson-Walker (FLRW) metric with signature $(-, +, +, +)$.

## 2 Inflationary Dynamics

In the simplest scenario, inflation is sourced by a single canonical scalar field $\phi$, known as the *inflaton field*, with a potential $V(\phi)$ which is minimally coupled to gravity. The system is described by the action

$$S[g_{\mu\nu}, \phi] = \int \mathrm{d}^4 x \, \sqrt{-g} \left( \frac{m_p^2}{2} R - \frac{1}{2} \partial_\mu \phi \, \partial_\nu \phi \, g^{\mu\nu} - V(\phi) \right). \tag{1}$$

The corresponding Lagrangian density $\mathcal{L}(\phi, \partial_\mu \phi)$ of the inflaton field is given by

$$\mathcal{L}(\phi, \partial_\mu \phi) = -\frac{1}{2} \partial_\mu \phi \, \partial^\mu \phi - V(\phi), \tag{2}$$

Varying (1) with respect to $\phi$ results in the field equation

$$\frac{\partial \mathcal{L}}{\partial \phi} - \left( \frac{1}{\sqrt{-g}} \right) \partial_\mu \left( \sqrt{-g} \, \frac{\partial \mathcal{L}}{\partial \left( \partial_\mu \phi \right)} \right) = 0. \tag{3}$$

At the background level, the system is described by a homogeneous inflaton condensate $\phi(t)$ in a spatially flat FRW universe with metric

$$\mathrm{d}s^2 = -\mathrm{d}t^2 + a^2(t)\left[\mathrm{d}x^2 + \mathrm{d}y^2 + \mathrm{d}z^2\right], \tag{4}$$

and energy-momentum tensor

$$T^{\mu}{}_{\nu} = \mathrm{diag}\left(-\rho_{\phi}, p_{\phi}, p_{\phi}, p_{\phi}\right). \tag{5}$$

The energy density, $\rho_{\phi}$, and pressure, $p_{\phi}$, of the inflaton energy-momentum tensor are given by

$$\rho_{\phi} = \frac{1}{2}\dot{\phi}^2 + V(\phi), \tag{6}$$

$$p_{\phi} = \frac{1}{2}\dot{\phi}^2 - V(\phi), \tag{7}$$

The evolution of the scale factor $a(t)$ is governed by the time-time and space-space components of Einstein's field equations, called the Friedmann equations, given by

$$\left(\frac{\dot{a}}{a}\right)^2 \equiv H^2 = \frac{1}{3m_p^2}\rho_{\phi} = \frac{1}{3m_p^2}\left[\frac{1}{2}\dot{\phi}^2 + V(\phi)\right], \tag{8}$$

$$\dot{H} \equiv \frac{\ddot{a}}{a} - H^2 = -\frac{1}{2m_p^2}\dot{\phi}^2, \tag{9}$$

where $H \equiv \dot{a}/a$ is the Hubble parameter and $\rho_{\phi}$ satisfies the conservation equation

$$\dot{\rho}_{\phi} = -3H\left(\rho_{\phi} + p_{\phi}\right), \tag{10}$$

which can be written as the equation of motion of the inflaton field

$$\ddot{\phi} + 3H\dot{\phi} + V_{,\phi}(\phi) = 0. \tag{11}$$

The evolution of various physical quantities during inflation is usually described with respect to the number of e-folds of expansion which is given by $N = \ln(a/a_i)$, where $a_i$ is an arbitrary epoch at early times during inflation (before the CMB pivot scale made its Hubble-exit). A better physical quantity to specify an epoch with scale factor $a$ during inflation is the *number of e-folds before the end of inflation* which is defined as

$$N_e(a) = \ln\left(\frac{a_e}{a}\right) = \int_t^{t_e} H(t)\,\mathrm{d}t, \tag{12}$$

where $H(t)$ is the Hubble parameter during inflation, and $a_e$ denotes the scale factor at the end of inflation, hence $N_e = 0$ corresponds to the end of inflation. Typically a period of almost exponential (quasi-de Sitter) inflation, lasting for at least 60-70 e-folds, is required to successfully address the fine tuning problems of the standard hot Big Bang model. We denote $N_*$ as the number of e-folds (before the end of inflation) when the CMB pivot scale $k_* = (aH)_* = 0.05\ \mathrm{Mpc}^{-1}$ made its Hubble-exit during inflation. While we have fixed $N_* = 60$ for the most part of this work, it is important to note that the exact value of $N_*$ depends upon the post-inflationary reheating history [16].

The quasi-exponential expansion during inflation usually corresponds to a slowly rolling inflaton field down its potential $V(\phi)$. For a wide variety of functional forms of the inflaton potential $V(\phi)$, there exists a *slow-roll regime* of inflation, enforced by the the Hubble friction term [20, 50] in Eq. (11). The slow-roll regime can be formally characterised by two (kinematic) Hubble slow-roll parameters $\epsilon_H$, $\eta_H$, defined as [7, 51, 52]

$$\epsilon_H = -\frac{\dot{H}}{H^2} = \frac{1}{2m_p^2}\frac{\dot{\phi}^2}{H^2}, \tag{13}$$

$$\eta_H = -\frac{\ddot{\phi}}{H\dot{\phi}} = \epsilon_H + \frac{1}{2\epsilon_H}\frac{d\epsilon_H}{dN_e}, \tag{14}$$

where the slow-roll conditions are given by

$$\epsilon_H,\ \eta_H \ll 1\,.\tag{15}$$

The slow-roll conditions (15) are often specified in terms of the (dynamical) potential slow-roll parameters [7], defined by

$$\epsilon_v = \frac{m_p^2}{2}\left(\frac{V_{,\phi}}{V}\right)^2\,,\quad \eta_v = m_p^2\left(\frac{V_{,\phi\phi}}{V}\right)\,.\tag{16}$$

When $\epsilon_H,\ \eta_H \ll 1$, we get $\epsilon_H \simeq \epsilon_v$ and $\eta_H \simeq \eta_v - \epsilon_v$. Since $H = \dot{a}/a$, from Eq. (13), we obtain

$$\frac{\ddot{a}}{a} = \left(1 - \epsilon_H\right)H^2\,,\tag{17}$$

which implies that expansion of space is accelerating, $\ddot{a} > 0$, if $\epsilon_H < 1$. For $\epsilon_H \ll 1$, acceleration of space becomes nearly exponential.

# 3 Quantum fluctuations during inflation

In order to study fluctuations around the background described in Sec. 2, we work in the framework of linear perturbation theory where we split the metric and inflaton field into their corresponding homogeneous background pieces and fluctuations, namely

$$g_{\mu\nu}(t,\vec{x}) = \bar{g}_{\mu\nu}(t) + \delta g_{\mu\nu}(t,\vec{x});\quad \varphi(t,\vec{x}) = \phi(t) + \delta\varphi(t,\vec{x})\,.$$

The perturbed metric $\delta g_{\mu\nu}$ has 10 degrees of freedom, out of which only two are independent, while the rest are fixed by gauge freedom, and the GR Hamiltonian and momentum constraints [53]. At linear order in perturbation theory, one gauge-invariant scalar degree of freedom (which is approximately massless during slow-roll inflation), and two gauge-invariant (transverse and traceless) massless tensor degrees of freedom are guaranteed to exist in the single-field inflationary paradigm [53, 54].

In linear perturbation theory, the dynamics of the perturbed fields are described by the corresponding quadratic actions. In the following, we first discuss the inflationary scalar fluctuations, before moving on to the tensor fluctuations.

**Scalar fluctuations during inflation**

The dynamics of gauge-invariant scalar fluctuations during inflation, known as the *curvature perturbations*[1] $\zeta$ is described in the comoving gauge [7, 54] by the quadratic Action

$$S_{(2)}[\zeta] = \frac{1}{2}\int \mathrm{d}\tau\mathrm{d}x^3\,z^2\left[(\zeta')^2 - (\partial_i\zeta)^2\right]\,.\tag{18}$$

It is more convenient to work with a canonical field variable $v$, known as the *Mukhanov-Sasaki variable*, which is defined as

$$v(\tau,\vec{x}) \equiv z\,\zeta(\tau,\vec{x});\quad \text{with}\quad z = am_p\sqrt{2\epsilon_H} = \frac{a\dot{\phi}}{H}\,.\tag{19}$$

The corresponding action for $v$ is given by

$$S_{(2)}[v] = \frac{1}{2}\int \mathrm{d}\tau\mathrm{d}x^3\left[(v')^2 - (\partial_i v)^2 + \frac{z''}{z}v^2\right]\tag{20}$$

---

[1]To be specific, the comoving curvature perturbation $\mathcal{R}$ is related to the curvature perturbation on uniform-density hypersurfaces, $\zeta$, and both are equal during slow-roll inflation and on super-Hubble scales, $k \ll aH$, in general (see Ref. [7]).

where the ($'$) denotes derivative with respect to conformal time $\tau = \int \frac{dt}{a(t)} \simeq \frac{-1}{aH}$ for quasi-de Sitter expansion. The Fourier modes $v_k$ of the Mukhanov-Sasaki field satisfy the *Mukhanov-Sasaki equation* [55, 56]

$$v_k'' + \left( k^2 - \frac{z''}{z} \right) v_k = 0 \, , \tag{21}$$

where the effective mass term is given by [52, 57]

$$\frac{z''}{z} = (aH)^2 \left( 2 - \epsilon_1 + \frac{3}{2}\epsilon_2 + \frac{1}{4}\epsilon_2^2 - \frac{1}{2}\epsilon_1\epsilon_2 + \frac{1}{2}\epsilon_2\epsilon_3 \right) , \tag{22}$$

$$\Rightarrow \frac{z''}{z} = (aH)^2 \left[ 2 + 2\epsilon_H - 3\eta_H + 2\epsilon_H^2 + \eta_H^2 - 3\epsilon_H\eta_H - \frac{1}{aH}\eta_H' \right] , \tag{23}$$

with $\epsilon_1 = \epsilon_H$ and where

$$\epsilon_{n+1} = -\frac{d \ln \epsilon_n}{dN_e} \tag{24}$$

are the 'Hubble flow' parameters. For a given Fourier mode $k$, at sufficiently early times when it is deep inside the Huuble radius, *i.e.* $k \gg aH$, the field $v$ is assumed to be in the Bunch-Davies vacuum state [58], for which its mode functions satisfy

$$v_k \to \frac{1}{\sqrt{2k}} e^{-ik\tau} \, . \tag{25}$$

During inflation as the comoving Hubble radius decreases, one by one different comoving modes become super-Hubble *i.e* $k \ll aH$. On super-Hubble scales, Eq. (21) dictates that $|v_k| \propto z$, which implies that $\zeta_k$ approaches a constant value. In fact by solving the Mukhanov-Sasaki Eq. (21) we can determine the (dimensionless) primordial power spectrum of $\zeta$ using the definition [53]

$$P_\zeta \equiv \frac{k^3}{2\pi^2} |\zeta_k|^2 \Big|_{k \ll aH} = \frac{k^3}{2\pi^2} \frac{|v_k|^2}{z^2} \Big|_{k \ll aH} . \tag{26}$$

In the slow-roll regime, solving the Mukhanov-Sasaki equation with suitable Bunch-Davies vacuum conditions leads to the slow-roll approximated expression for the scalar power spectrum as [7]

$$P_\zeta(k) = \frac{1}{8\pi^2} \left( \frac{H}{m_p} \right)^2 \frac{1}{\epsilon_H} , \tag{27}$$

where, $H$ and $\epsilon_H$ appearing in the right-hand side of the above equation should be calculated at the time of Hubble-exit of the mode $k$, namely, when $k = aH$. Note that one can also directly solve for the Fourier modes of the comoving curvature perturbation $\zeta$ (instead of the Mukhanov-Sasaki variable $v$, see Sec. 6) which satisfies the equation

$$\zeta_k'' + 2 \left( \frac{z'}{z} \right) \zeta_k' + k^2 \zeta_k = 0 \tag{28}$$

and impose the corresponding Bunch-Davies initial conditions for $\zeta_k$. The friction term in Eqn. (28) is given by

$$\frac{z'}{z} = aH (1 + \epsilon_H - \eta_H). \tag{29}$$

Before moving forward, we stress that the slow-roll regime of inflation necessarily requires both the slow-roll parameters to be small *i.e.* $\epsilon_H \ll 1$ and $\eta_H \ll 1$. Violation of either of these conditions invalidates the above analytical treatment. When either of the slow-roll conditions is violated, which is the situation in the context of primordial black hole formation, a more accurate determination of $P_\zeta$ is provided by solving the Mukhanov-Sasaki Eqn. (21) numerically. In fact, the computation of power spectrum when the slow-roll approximation is violated will be our primary focus in this work.

**Tensor fluctuations**

The gauge-invariant tensor fluctuations $h_{ij}(\tau, \vec{x})$ during inflation are described by the quadratic action [53, 54]

$$S^{(2)}[h_{ij}] = \frac{1}{2} \int d\tau \, d^3\vec{x} \left( \frac{am_p}{2} \right)^2 \left[ h_{ij}^{\prime 2} - (\partial_l h_{ij})^2 \right], \tag{30}$$

where the transverse and traceless tensor field $h_{ij}(\tau, \vec{x})$ satisfies

$$\partial^i h_{ij} = 0; \quad h_i^i = 0, \tag{31}$$

and can be decomposed into its two orthogonal polarization components[2],

$$h_{ij}(\tau, \vec{x}) = \frac{1}{\sqrt{2}} \begin{bmatrix} h_+ & h_\times & 0 \\ h_\times & -h_+ & 0 \\ 0 & 0 & 0 \end{bmatrix} = \frac{1}{\sqrt{2}} \begin{bmatrix} 1 & 0 & 0 \\ 0 & -1 & 0 \\ 0 & 0 & 0 \end{bmatrix} h_+(\tau, \vec{x}) + \frac{1}{\sqrt{2}} \begin{bmatrix} 0 & 1 & 0 \\ 1 & 0 & 0 \\ 0 & 0 & 0 \end{bmatrix} h_\times(\tau, \vec{x}) \tag{32}$$

$$= \frac{1}{\sqrt{2}} \epsilon_{ij}^+ h_+(\tau, \vec{x}) + \frac{1}{\sqrt{2}} \epsilon_{ij}^\times h_\times(\tau, \vec{x}), \tag{33}$$

with $\epsilon_{ij}^\lambda \epsilon^{ij\,\lambda'} = 2\delta^{\lambda\lambda'}$; $\lambda = \{+, \times\}$. The action (30) can be rewritten as

$$S^{(2)}[h_+, h_\times] = \frac{1}{2} \int d\tau \, d^3\vec{x} \left( \frac{am_p}{2} \right)^2 \sum_{\lambda=+,\times} \left[ (h_\lambda{}')^2 - (\partial_l h_\lambda)^2 \right]. \tag{34}$$

The evolution equation for the Fourier modes of the tensor fluctuations is given by

$$\left( h_k^\lambda \right)'' + 2 \left( \frac{a'}{a} \right) \left( h_k^\lambda \right)' + k^2 h_k^\lambda = 0. \tag{35}$$

We define the two Mukhanov-Sasaki variables for the tensor perturbations to be

$$v_\lambda = \left( \frac{am_p}{2} \right) h_\lambda, \tag{36}$$

in terms of which the action (34) takes the form

$$S^{(2)}[v_+, v_\times] = \frac{1}{2} \int d\tau \, d^3\vec{x} \sum_{\lambda=+,\times} \left[ (v_\lambda{}')^2 - (\partial_i v_\lambda)^2 + \frac{a''}{a} v_\lambda{}^2 \right], \tag{37}$$

which is equivalent to the action of two massless scalar fields in quasi-de Sitter spacetime, as can be inferred by comparing this with Eq. (20). The power spectrum of tensor fluctuations, defined by

$$\mathcal{P}_T(k) = \frac{k^3}{2\pi^2} \left( |h_+|^2 + |h_\times|^2 \right) = \frac{k^3}{2\pi^2} \left( \frac{2}{am_p} \right)^2 \left( |v_+|^2 + |v_\times|^2 \right), \tag{38}$$

on super-Hubble scales $|k\tau| \to 0$ becomes [7, 53]

$$\mathcal{P}_T(k) \Big|_{k \ll aH} = \frac{2}{\pi^2} \left( \frac{H}{m_p} \right)^2. \tag{39}$$

Note that, unlike the quadratic action. (18) for the scalar fluctuations, the tensor action (30) does not depend upon $z$, rather it depends only upon the scale factor $a$. Hence, as long as the quasi-de Sitter approximation is valid, $i.e$ $\epsilon_H \ll 1$, power spectrum of tensor fluctuations does not get affected by an appreciable amount when slow-roll conditions are is violated. However, this statement is true only at the linear order in perturbation theory. Tensor fluctuations at the second order in perturbation theory can be sourced (induced) by the first-order scalar fluctuations [59–63] (also see Ref. [64] and references therein).

---

[2]Note that in Eq. (32) we have assumed the tensor modes to be propagating along the $z$-direction, $i.e.$ along $(0, 0, 1)$; while Eq. (33) is valid in general, independent of the aforementioned assumption.

## 3.1  Primordial fluctuations at large cosmological scales

On large cosmological scales which are accessible to CMB observations, the scalar and tensor power spectra are typically approximated to be of the power-law form, represented by

$$P_\zeta(k) \; = \; A_s \left( \frac{k}{k_*} \right)^{n_s - 1} ; \tag{40}$$

$$P_T(k) \; = \; A_T \left( \frac{k}{k_*} \right)^{n_T} ; \tag{41}$$

where the amplitudes of the scalar and tensor power spectra at the pivot scale $k = k_*$ are given by[3]

$$A_s \equiv P_\zeta(k_*) \; = \; \frac{1}{8\pi^2} \left( \frac{H}{m_p} \right)^2 \frac{1}{\epsilon_H} \Bigg|_{\phi = \phi_*} ; \tag{42}$$

$$A_T \equiv P_T(k_*) \; = \; \frac{2}{\pi^2} \left( \frac{H}{m_p} \right)^2 \Bigg|_{\phi = \phi_*} . \tag{43}$$

Where $\phi_*$ is the value of the inflaton field at the epoch of Hubble exit of the CMB pivot scale $k_*$. The scalar and tensor spectral indices in the slow-roll regime are given by [7]

$$n_s - 1 \equiv \frac{d \ln P_\zeta}{d \ln k} \; = \; 2\eta_H - 4\epsilon_H ; \tag{44}$$

$$n_T \equiv \frac{d \ln P_T}{d \ln k} \; = \; -2\epsilon_H . \tag{45}$$

Accordingly, the tensor-to-scalar ratio $r$ is defined by

$$r \equiv \frac{A_T}{A_s} = 16\,\epsilon_H , \tag{46}$$

leading to the single field consistency relation

$$r = -8\,n_T . \tag{47}$$

which can be used as a smoking-gun test for the slow-roll inflationary paradigm of a single scalar field with a canonical kinetic term[4], see Refs. [7, 65]. From the aforementioned analysis, it is clear that the slow-roll parameters $\epsilon_H$ and $\eta_H$ play a pivotal role in characterising the power spectra of scalar and tensor fluctuations during inflation.

Observations made by various CMB missions, such as the Planck team [15] and the BICEP/Keck collaboration, we have imposed stringent constraints on the nature of the primordial fluctuations. In particular, the latest data release from Planck 2018 [15] constrain the amplitide of scalar fluctuations at the CMB pivot scale $k_* = 0.05 \; \text{Mpc}^{-1}$ to be

$$A_s = 2.1 \times 10^{-9} , \tag{48}$$

while the $2\sigma$ constraint on the scalar spectral index is given by

$$n_s \in [0.957, 0.976] . \tag{49}$$

Similarly, the upper bound on the tensor-to-scalar ratio $r$, from the combined analysis of Planck 2018 [15] and BICEP/Keck [17], is found to be

$$r \le 0.036 , \tag{50}$$

---

[3]In general, $k$ may correspond to any scale in the range $k \in [0.0005, 0.5] \; \text{Mpc}^{-1}$ accessible to the CMb observations. However, in order to derive stringent constraints on the inflationary observables $\{n_s, r\}$, we primarily focus on the CMB pivot scale, *i.e.* $k \equiv k_* = 0.05 \; \text{Mpc}^{-1}$.

[4]Note that the consistency relation does not hold for a non-canonical scalar field for which the speed of sound $c_s^2 \ne 1$, as well as for multi-filed inflation, in general.

which further translates into $A_T \leq 3.6 \times 10^{-2} A_S$. This imposes an upper bound on the inflationary Hubble scale $H^{\text{inf}}$ to be

$$H^{\text{inf}} \leq 4.7 \times 10^{13} \text{ GeV}, \tag{51}$$

Similarly the CMB bound on $r$, when combined with Eqs. (46), (49) and (44), translates into the corresponding constraints on the slow-roll parameters [19]

$$\epsilon_H \leq 0.00225 \, ; \tag{52}$$

$$|n_T| \leq 0.0045 \, ; \tag{53}$$

$$|\eta_H| \in [0.0075, 0.0215] \, . \tag{54}$$

Furthermore, the equation of state (EoS) $w_\phi$ of the inflaton field, defined as

$$w_\phi = \frac{\frac{1}{2}\dot{\phi}^2 - V(\phi)}{\frac{1}{2}\dot{\phi}^2 + V(\phi)} = -1 + \frac{2}{3}\epsilon_H(\phi), \tag{55}$$

is constrained, during slow-roll inflation, to be

$$w_\phi \leq -0.9985 \, , \tag{56}$$

which demonstrates that the expansion of the universe during inflation was nearly exponential (quasi-de Sitter) and provides strong support for the single-field slow-roll paradigm of inflation on large cosmological scales. In fact, for simple slow-roll potentials that do not possess any features on small scales outside the CMB window, the latest CMB observations favour asymptotically-flat concave potentials (featuring either one or two plateau wings) with $n_S \simeq 0.965$ and $r \leq 0.036$ over their convex counterparts, see Ref. [19] for a detailed discussion. Given that power spectrum is nearly scale-invariant with a small red tilt, *i.e.* $n_S - 1 \lesssim 0$, large-scale primordial fluctuations are more important in the slow-roll paradigm and nothing drastic is expected to happen on smaller cosmological scales as long as they are super-Hubble at the end of inflation. However, the same may not be true for small-scale fluctuations that are inaccessible to the CMb observations.

Before proceeding further to discuss small-scale inflationary fluctuations, it is important to make the following nomenclature concrete and explicit (to be consistent with the standard nomenclature in the inflationary literature).

- *Quasi-de Sitter* expansion corresponds to the condition $\epsilon_H \ll 1$.

- *Slow-roll* inflation corresponds to $\epsilon_H \ll 1$ and $|\eta_H| \ll 1$.

Under either of the aforementioned assumptions, the conformal time during inflation is given by

$$-\tau \simeq \frac{1}{aH} \, . \tag{57}$$

## 3.2 Small-scale primordial fluctuations

Our discussion in Sec. 3.1 established that the latest CMB observations point towards nearly scale-invariant primordial scalar fluctuations with relatively low tensor-to-scalar ratio. Furthermore, the current observational constraints [66] are consistent with predominantly Gaussian primordial fluctuations. Within the canonical single-field inflationary paradigm, this provides support for slow-roll inflation with an asymptotically-flat potential corresponding to the field space when the observable CMB scales made their Hubble exit during inflation. However, the current CMB and LSS observations probe only about 7-8 e-folds of inflation around the Hubble-exit time of the CMB pivot scale. In particular, the CMB observations probe

the primordial fluctuations with comoving scales $k_{\text{CMB}} \in [0.0005, 0.5]$ Mpc$^{-1}$ (including the pivot scale $k_* = 0.05$ Mpc$^{-1}$) corresponding to the angular multipoles $l \in [2, 2500]$ in the sky. Additionally, Lyman-$\alpha$ forest observations impose constraints on the primordial power spectrum upto $k \simeq \mathcal{O}(1)$ Mpc$^{-1}$ (see Ref. [29]).

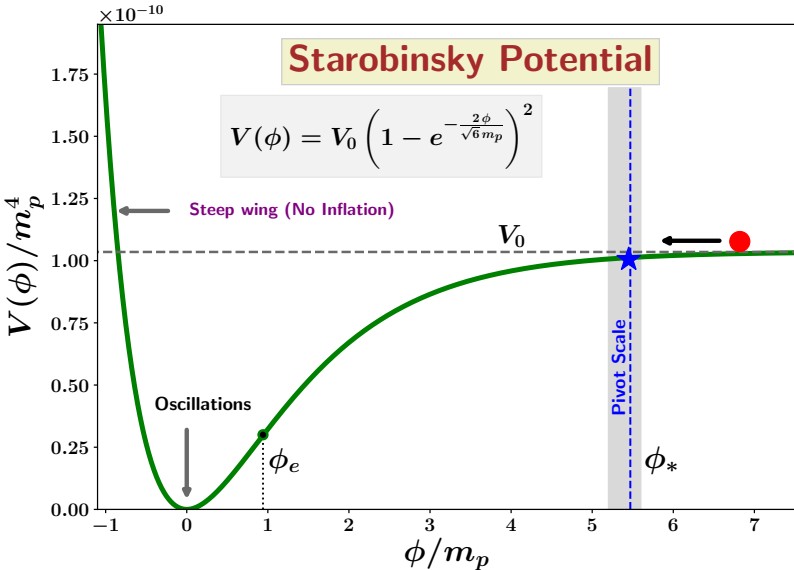

Figure 1: The Starobinsky potential (58) is plotted in green curve with the CMB pivot scale $k_* = 0.05$ Mpc$^{-1}$ being marked by a blue star as well as the CMB window $k_{\text{CMB}} \in [0.0005, 0.5]$ Mpc$^{-1}$ shown in grey shade in the field space. The CMB window constitutes only a small portion of the field space explored by the inflaton in between $\phi_{\text{CMB}}$ and end of inflation $\phi_e$.

A large portion of the inflationary evolution which accounts roughly to ($\geq$) 50 e-folds of expansion, corresponding to length scales that are much smaller than those probed by the CMB, remains inaccessible to the CMB and large scale structure observations at present. Consequently, the associated dynamics of the inflaton field also remains unexplored. For example, Fig. 1 demonstrates that the CMB window constitutes only a small portion of the observationally available field space between the Hubble-exit of the largest scales in the sky and the smallest scale at the end of inflation for the Starobinsky potential [1, 67]

$$V(\phi) = V_0 \left( 1 - e^{-\frac{2}{\sqrt{6}} \frac{\phi}{m_p}} \right)^2. \tag{58}$$

A significant departure from the slow-roll regime $\epsilon_H, \eta_H \ll 1$, which may be induced by a change in the inflaton dynamics, would lead to interesting observational signatures on small scales. In particular, if the inflaton potential exhibits a broad (near) inflection point-like feature at an intermediate field range $\phi \in (\phi_e, \phi_*)$, then the scalar power spectrum corresponding to scales that become super-Hubble around the time when the inflaton rolls through the feature, might get amplified significantly during inflation, and lead to the formation of Primordial Black Holes (PBHs) upon their Hubble re-entry during the hot Big Bang phase.

A variety of possible small-scale features have been proposed in the recent literature which can induce a significant departure from the standard scale-invariant power spectrum, see Ref. [68]. However, in this work we will primarily focus on potentials with a tiny local bump/dip like feature [44] in order to illustrate the efficiency of our numerical analysis. Our code can be used to simulate a number of different types of features in the inflaton potential.

PBH formation requires the amplification of the inflationary power spectrum by roughly a factor of $10^7$ within less than 40 e-folds of expansion (on smaller primordial scales, $k \gg k_*$) as illustrated in Fig. 3.

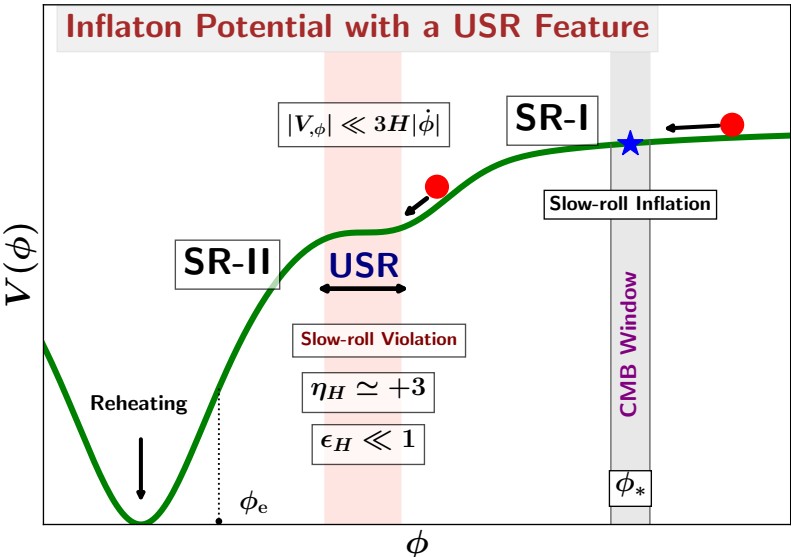

Figure 2: A schematic illustration of an inflationary plateau potential (in solid green curve). The 'CMB Window' corresponds to field values associated with the Hubble-exit epochs of comoving scales $k_{\mathrm{CMB}} \in [0.0005, 0.5]$ Mpc$^{-1}$ that are accessible to the latest CMB observations. The potential exhibits a (broad) flat inflection point-like small-scale feature (shown in the salmon shading) which leads to an ultra slow-roll (USR) phase of inflation. As the inflaton enters into the transient USR phase following the standard CMB scale slow-roll (SR-I) regime, the second slow-roll condition is violated, namely $\eta_H \simeq +3$. Subsequently, the inflaton exits the USR regime to enter into another slow-roll phase (SR-II) before the end of inflation.

Therefore the quantity $|\Delta \ln \epsilon_H|/\Delta N$, and hence also $|\eta_H|$, can grow to become of order unity, which violates the second slow-roll condition, as originally pointed out in Ref. [69]. In particular, the second Hubble slow-roll parameter $|\eta_H|$ becomes larger than unity, even though the first slow-roll parameter $\epsilon_H$ remains to be much smaller than unity. As a result, we can not use the slow-roll approximated formula in Eq. (42) to compute the scalar power spectrum. When slow-roll is violated, one must determine $P_\zeta$ by numerically integrating[5] the Mukhanov-Sasaki Eq. (21). We proceed as follows. We first discuss the simulations of inflationary background dynamics in section 4, where we also discuss how to generate phase-space portrait $\{\phi, \dot{\phi}\}$ during inflation. In section 5.1, we introduce our numerical scheme for studying quantum fluctuations during slow-roll inflation. We work with convex as well as asymptotically-flat potentials. In section 5.2, we apply our numerical scheme to potentials featuring a local bump/dip like feature that facilitates the amplification of scalar power on small primordial scales. We demonstrate that slow-roll formula (42) underestimates both the location as well as the height of scalar power spectrum $\mathcal{P}_\zeta(k)$ for both type of aforementioned features and hence one must solve the Mukhanov-Sasaki Eqn. (21) numerically to estimate the power spectrum accurately. We also demonstrate that the growth of the power spectrum obeys the steepest growth bounds discussed in [73, 76–78].

---

[5]For analytical approach, see Refs. [70–75].

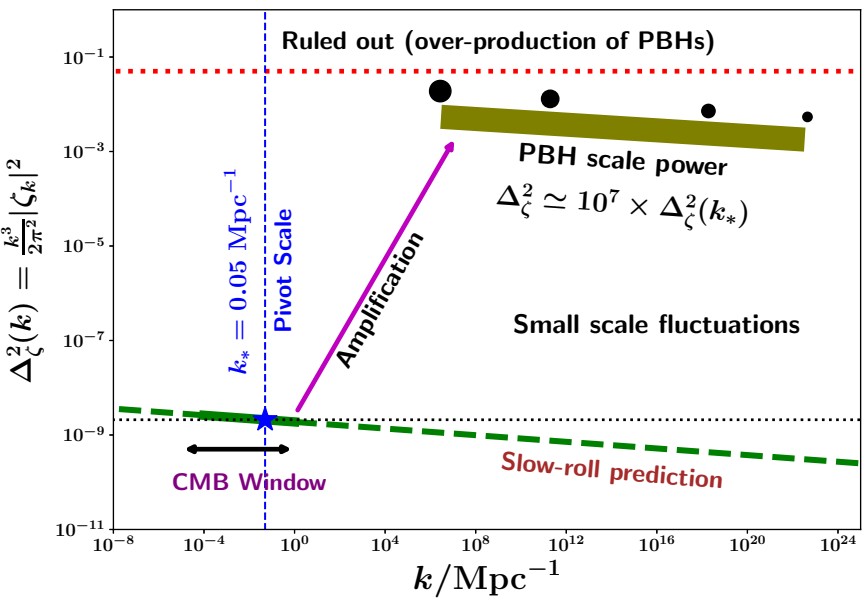

Figure 3: This figure shows the typical amplification of inflationary scalar power spectrum at smaller length scales required for PBH formation.

## 4 Numerical analysis of inflationary background dynamics

A complete analysis of the inflationary background dynamics can be obtained from the evolution of $\phi$, $\dot{\phi}$ and $H$. All of these quantities can be simulated by numerically solving equations (8), (9) and (11). The evolution of the scale factor follows directly from $H = \dot{a}/a$. Our system is defined by the following set of equations (as a function of cosmic time $t$)

$$H^2 = \frac{1}{3m_p^2}\left[\frac{1}{2}\dot{\phi}^2 + V(\phi)\right], \tag{59}$$

$$\dot{H} = -\frac{\dot{\phi}^2}{2m_p}, \tag{60}$$

$$\ddot{\phi} = -3H\dot{\phi} - V_{,\phi}(\phi), \tag{61}$$

where the functional form of the potential $V(\phi)$ is given by the specific inflationary model. However, the rest of the algorithm is largely model-independent. We can re-write the potential as

$$V(\phi) = V_0 f(\phi). \tag{62}$$

In order to carry out numerical simulations, it is convenient to write down the dynamical equations in terms of dimensionless variables (which also ensures that we do not need to worry about keeping track of units). Furthermore, it is important to re-scale the time variable by a factor $S$ which can be suitably chosen according to the energy scale of the dynamics[6]. Our primary dimensionless variables are defined as

---

[6]Depending upon the potential, we usually choose the value of $S$ to be in the range $S \in [10^{-5}, 10^{-3}]$.

$$T = \left( t\, m_p \right) S \,, \tag{63}$$

$$x = \frac{\phi}{m_p} \,, \tag{64}$$

$$y = \left( \frac{\dot{\phi}}{m_p^2} \right) \frac{1}{S} \,, \tag{65}$$

$$z = \left( \frac{H}{m_p} \right) \frac{1}{S} \,, \tag{66}$$

$$A = \left( a\, m_p \right) S \,. \tag{67}$$

In terms of these variables, the dynamical equations (to be simulated) take the form

$$\frac{\mathrm{d}x}{\mathrm{d}T} = y \,, \tag{68}$$

$$\frac{\mathrm{d}y}{\mathrm{d}T} = -3z\, y - \frac{v_0}{S^2} f_{,x}(x) \,, \tag{69}$$

$$\frac{\mathrm{d}z}{\mathrm{d}T} = -\frac{1}{2} y^2 \,, \tag{70}$$

$$\frac{\mathrm{d}A}{\mathrm{d}T} = A z \,. \tag{71}$$

We can also define the dimensionless potential to be

$$\frac{V(\phi)}{m_p^4} \equiv v_0 f(x) = \frac{V_0}{m_p^4} f(x) \,. \tag{72}$$

We can solve the aforementioned set of equations with appropriate initial conditions. In our analysis, we use the `odeint` function provided in the `scipy.integration` package. By incorporating initial conditions $\{x_i, y_i, z_i, A_i\}$ for the primary dynamical variables $\{x, y, z, A\}$, we simulate their time evolution during inflation. Accordingly, we determine the crucial derived/secondary (dimensionless) dynamical variables from the primary ones by[7]

$$N = \log \frac{A}{A_i} \tag{73}$$

$$\epsilon_H = \frac{1}{2} \frac{y^2}{z^2} \,, \qquad\qquad \eta_H = -\frac{1}{yz} \frac{\mathrm{d}y}{\mathrm{d}T} \,, \tag{74}$$

$$A_s = \frac{1}{8\pi^2} \frac{(Sz)^2}{\epsilon_H} \,, \qquad\qquad A_T = \frac{2}{\pi^2} (Sz)^2 \,, \tag{75}$$

$$n_s = 1 + 2\eta_H - 4\epsilon_H \,, \qquad\qquad n_T = -2\epsilon_H \,, \tag{76}$$

$$r = 16\epsilon_H \,, \tag{77}$$

where the last three Eqs. are valid only under the slow-roll approximations, as discussed in Sec. (4.2). We define $N_T$ to be the number of e-folds of accelerated expansion realised in between an arbitrary initial time and the end of inflation, which is marked by $\epsilon_H = 1$. We then define the more important quantity

---

[7]Note that the observables $A_s, A_T, n_s, n_T$, and $r$ are related to inflationary scalar and tensor fluctuations and the expressions given here are under slow-roll approximations, *i.e* $\epsilon_H, |\eta_H| \ll 1$, during which they can be determined purely from the dynamics of background quantities such as $H, \epsilon_H, \eta_H$. Computation of inflationary power spectra when slow-roll is violated is described in section 5.2.

$N_e = N_T - N$ as the instantaneous number of e-folds left before the end of inflation. Note that $N_e = 0$ at the end of inflation while $N_e > 0$ at early times. This will be our primary time variable against which we will be plotting the dynamics of different inflationary observables. In order to realise adequate amount of inflation, *i.e.* $N_T > 60$, initial value of scalar field must be large enough (and is model dependent). For most large field potentials, this value is of the order $\phi_i \lesssim \mathcal{O}(10)\, m_p$.

Composing the code involves typing down the dimensionless equations in the appropriate syntax and solving them by using an ODE solver. See our supplementary Python code [8] for details. For a particular model of interest, we need to input the inflaton potential in the form $\frac{V(\phi)}{m_p^4} = v_0 f(x)$. Since the slow-roll parameters and the duration $N_T$ do not strongly depend on $v_0$, we can initially set its value to roughly $v_0 = 10^{-10}$. We can later adjust the value of $v_0$ to yield the correct CMB normalised value of scalar power spectrum (48) at the pivot scale $N = N_*$. One can proceed in the following step-by-step algorithm.

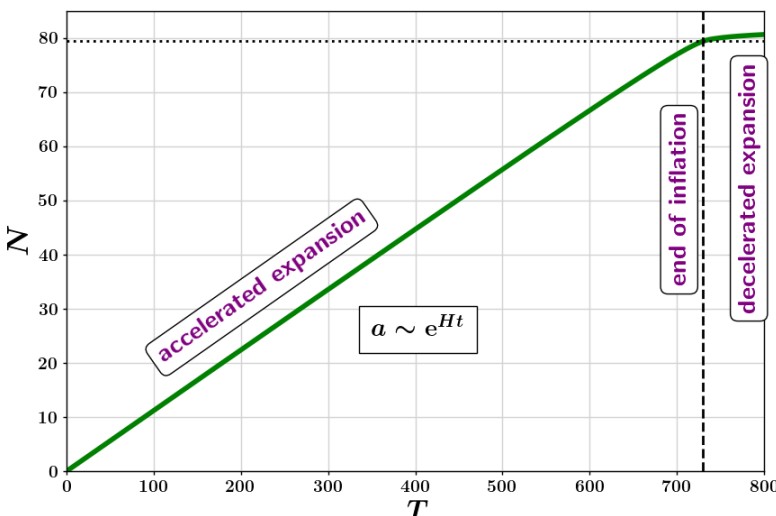

Figure 4: Time evolution of the number of e-folds (scale factor in the logarithm scale) of expansion of the universe is shown for Starobinsky potential (58). For the most part of inflation, the expansion is almost exponential (quasi-de Sitter) *i.e* $a \sim e^{Ht}$, leading to a rapid growth in the number of e-folds within a small amount of time. While after the end of inflation, the expansion is decelerated, leading to a much slower growth in the scale factor.

1. After setting the parameters of the potential and defining the function $f(x)$, we need to incorporate initial conditions for the four primary variables $\{x, y, z, A\}$. We enter appropriate initial conditions $x_i$, $y_i$ and $A_i$ in the following way. $A_i$ can be set arbitrarily in a spatially flat universe, however depending upon the energy scale of inflation, one can provide an appropriate value. We suggest a typical $A_i = 1 \times 10^{-3}$, although its precise value does not affect the dynamics. In regard to the initial value of $x$, we need to ensure that $x_i$ is large enough (or small enough if we are working with symmetry breaking hilltop type potentials) to yield adequate amount of inflation, *i.e* $N_T \geq 70$. As mentioned before, the typical value for large field models is $x_i \lesssim \mathcal{O}(10)$.

   Since we will be mostly working with potentials that exhibit slow-roll behaviour at initial times, and given that slow-roll trajectory is an attractor in relatively large field models [20], we can safely set $y_i = 0$, as long as $x_i$ is large enough. One can also incorporate slow-roll initial conditions from the

[8]https://github.com/bhattsiddharth/NumDynInflation

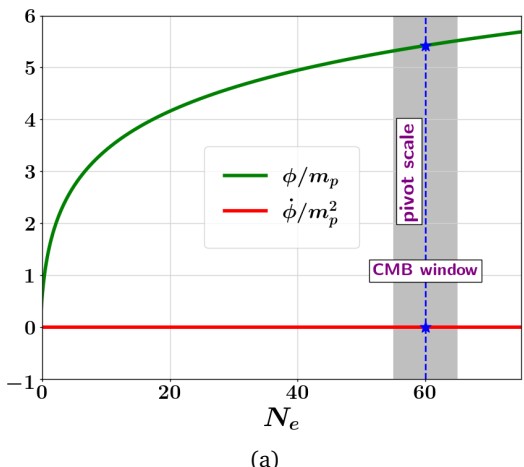
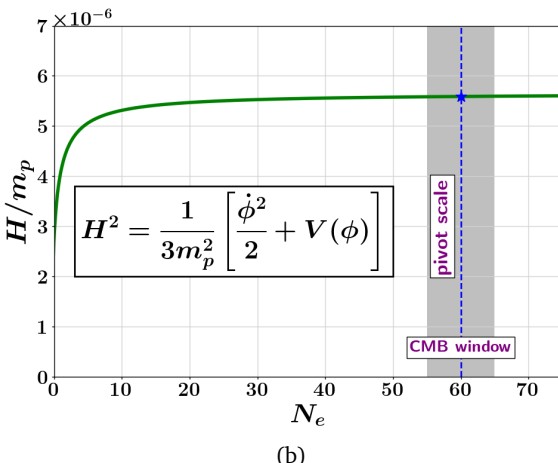

(a)                (b)

Figure 5: This figure describes the evolution of inflaton field $\phi$, and its speed $\dot{\phi}$ in the **left panel**, while the Hubble parameter $H$ in the **right panel** as a function of the number of e-folds before the end of inflation $N_e$ for Starobinsky potential (58). Note that during slow-roll inflation, $\dot{\phi}$ and $H$ are nearly constant, while $\phi$ changes quite slowly. However, $\phi$ and $H$ begin to change rapidly towards the end of inflation. After inflation ends, $\phi$ and $\dot{\phi}$ start oscillating around the minimum of the potential (which is not shown in this figure).

beginning, namely $y_i = -\frac{v_0}{3} \frac{f_{,x}}{Sz}$, as is usually done in practice[9]. Finally, the initial value of $z$ can be incorporated in terms of $x_i$, $y_i$ using the dimensionless Friedmann equation

$$z_i = \sqrt{\frac{1}{6} y_i^2 + \frac{1}{3} v_0 f(x_i) \frac{1}{S^2}}. \tag{78}$$

2. We then proceed to solve the system of equations by taking adequately small time steps $T$ in the appropriate range $T \in [T_i = 0, T_f]$. We then plot $N$ vs $T$ as given in Fig. 4 for Starobinsky potential. Typically, $N$ grows linearly with $T$ during near exponential inflation and a substantial decrease in the rate of growth of $N$ indicates the end of inflation.

3. In order to concretely determine the value of $N_T$, we plot $\epsilon_H$ vs $N$, and note the value of $N$ after which $\epsilon_H \geq 1$. By definition, initially $N = 0$. If $N_T < 70$, then we repeat this step by increasing the value of $x_i$, until we get $N_T \geq 70$. This is so that the CMB scale (corresponding to $N_e \simeq 60$ sits comfortably in our purview. (Alternatively, if inflation has not ended[10], *i.e* $\epsilon_H < 1$, at the end of our simulation, then either one can increase the value of $T_f$ or decrease $x_i$. We suggest the latter.) We can then define the number of e-folds before the end of inflation to be $N_e = N_T - N$.

4. The pivot scale can then be fixed to an appropriate value, for example $N_e = 60$, as used in this work[11]. Fig. 5 describes the evolution of $\phi$, $\dot{\phi}$, and $H$, while Fig. 6 illustrates the dynamics of slow-

---

[9]Note that for phase-space analysis, we need to incorporate arbitrary values of $x_i$, $y_i$ (consistent with fixed initial $z_i$) which may be away from the slow-roll trajectory as discuss below.

[10]Note that if one simulates the cosmological equations in terms of number of e-folds $N$, rather than cosmic time $t$, this step can usually be avoided by simulating the system from $N = 0$ to $N = 70$. However, one has to adjust the value of $x_i$ in order to get enough inflation.

[11]We again stress that the exact value of $N_e$ depends upon the reheating history in the post-inflationary universe. While we fix it to $N_e = 60$ for the purpose of illustration, in principle, our numerical framework allows for incorporating a different value of $N_e$ without any trouble.

roll parameters $\epsilon_H$, $|\eta_H|$ for Starobinsky potential, as determined from our code. As mentioned before, we usually plot the dynamics of inflation as a function of $N_e$.

5. In order to accurately fix the value of $v_0$, we need to impose $A_S = 2.1 \times 10^{-9}$ at the pivot scale $N_e = 60$. If $A_S$ is lower than expected for the given value of $v_0$, then we increase the value of $v_0$ or vice versa until we arrive at the correct value of $A_S$, and fix the corresponding value of $v_0$.

Following the aforementioned algorithm, we can easily simulate the inflationary background dynamics and investigate the evolution of relevant quantities of our interest. Before going forward, we would like to stress that many of the aforementioned steps (in the present version of our code) are rather meant to be carried out manually by the user. While we are already developing an automated version of this code (which will be presented in the revised version of our paper), we believe that the present version will help the user to understand the inflationary dynamics much better.

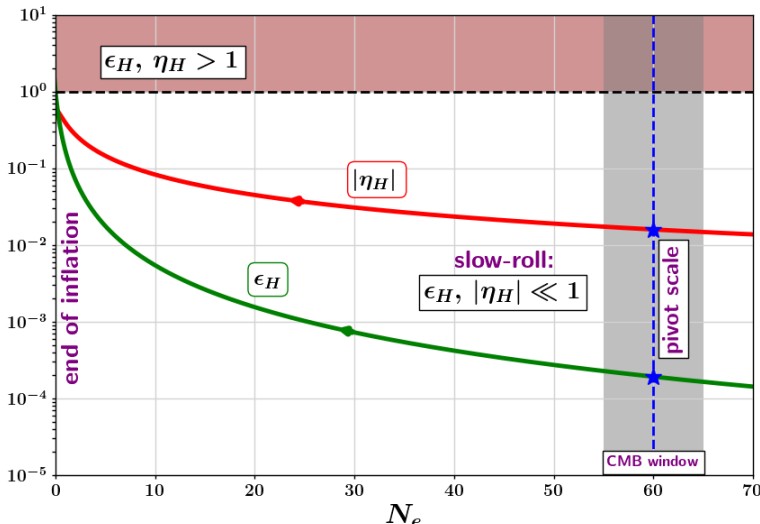

Figure 6: Evolution of the slow-roll parameters $\epsilon_H$ and $\eta_H$ is shown as a function of the number of e-folds before the end of inflation $N_e$ for Starobinsky potential (58). From this plot, it is easy to notice that at early times when $N_e \gg 1$, the slow-roll conditions are satisfied *i.e* $\epsilon_H$, $|\eta_H| \ll 1$. However, the slow-roll conditions are violated towards the end of inflation (marked by $N_e = 0$ and $\epsilon_H = 1$).

## 4.1 Phase-space analysis

Phase-space analysis of inflationary dynamics is usually carried out to determine the set of initial conditions that results in adequate amount of inflation, and hence it is important to assess the generality of initial conditions for inflation [20, 50, 79]. For a spatially flat background, the phase-space portrait consists of trajectories of $\{\phi, \dot{\phi}\}$ for different initial conditions, with fixed $H_i$. The standard algorithm to generate such a plot is the following.

1. The initial energy scale of inflation is kept constant by fixing the value of initial Hubble parameter in the phase-space portrait simulations. A typical value often used is $H_i \leq m_p$ (see Ref. [20] for detail). Hence, the user is expected to incorporate an appropriate value of $z_i$.

2. One can then input a suitable value of $x_i$ and determine the value of $y_i$ for a given potential function $f(x)$ from the dimensionless Friedmann Eqn. (78) as

$$y_i = \pm\sqrt{6}\sqrt{z_i^2 - \frac{1}{3}v_0 f(x_i)\frac{1}{S^2}}\,. \tag{79}$$

3. With these initial conditions, one can then simulate the system of dimensionless differential equations for $\{x, y, z, A\}$ from $T_i = 0$ till an appropriate $T_f$. One can then repeat the same step by incorporating a number of different values of of $x_i$ in order to generate the phase-space portrait for the given potential. We provide the GitHub link to our phase-space portrait framework here [12]. The phase-space portraits for Starobinsky potential (58), and quadratic potential $V(\phi) \propto \phi^2$ are illustrated in the left and right panels of Fig. 7 respectively.

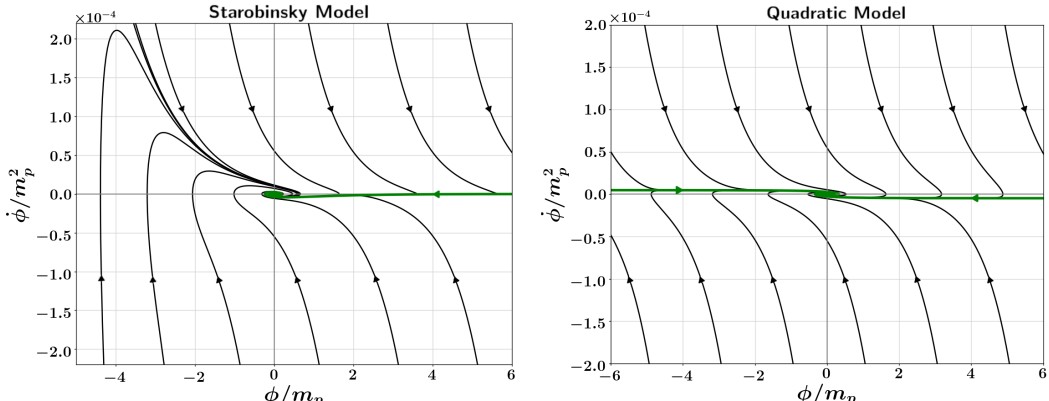

Figure 7: The phase-space portrait $\{\phi, \dot{\phi}\}$ of the inflaton field has been illustrated for Starobinsky potential (58) in the **left panel**, and for quadratic potential $V(\phi) = \frac{1}{2}m^2\phi^2$ in the **right panel** corresponding to different initial conditions $\{\phi_i, \dot{\phi}_i\}$ (plotted in solid black colour) with a fixed initial scale $H_i$. The figure demonstrates that trajectories commencing from a large class of initial field values (including those with large initial velocities $\dot{\phi}_i$) quickly converge towards the slow-roll attractor separatrix $\dot{\phi} = -V_{,\phi}/3H \simeq$ const. (plotted in green colour) as can be seen from the rapid decline in the inflaton speed until they meet the green colour curve. After the end of inflation, the inflaton begins to oscillate around the minimum of the potential.

In order to determine the degree of generality of inflation, we need to define a measure for the distribution of $\{\phi_i, \dot{\phi}_i\}$. The correct choice for the measure might depend on the quantum theory of gravity. However, a uniform measure is usually considered in the literature. Interested readers are referred to [20, 79] for further detail.

## 4.2 Quantum fluctuations under the slow-roll approximation

In section 3, we described the expressions for a number of inflationary observables such as $A_s$, $A_T$, $n_s$, $n_T$, and $r$ associated with the scalar and tensor power spectra which can be determined purely from the dynamics of background quantities such as $H$, $\epsilon_H$, $\eta_H$ under the slow-roll approximation. Hence they can be conveniently determined from our background dynamics code as discussed earlier in section 4. For example, the scalar and tensor power spectra for Starobinsky inflation have been plotted in Fig. 8 as a function $N_e$. Similarly, one can plot the spectral indices[13] $n_s - 1$ and $n_T$ and determine their values at the pivot scale $N_*$. The spectral indices for Starobinsky potential have been plotted in Fig. 9.

---

[12]https://github.com/bhattsiddharth/NumDynInflation/blob/main/inf_dyn_phase.py

[13]In the standard literature, one usually plots $r$ vs $n_s$ for a given inflaton potential for a range of possible values of $N_* \in [50, 60]$ which can also be done easily using our code.

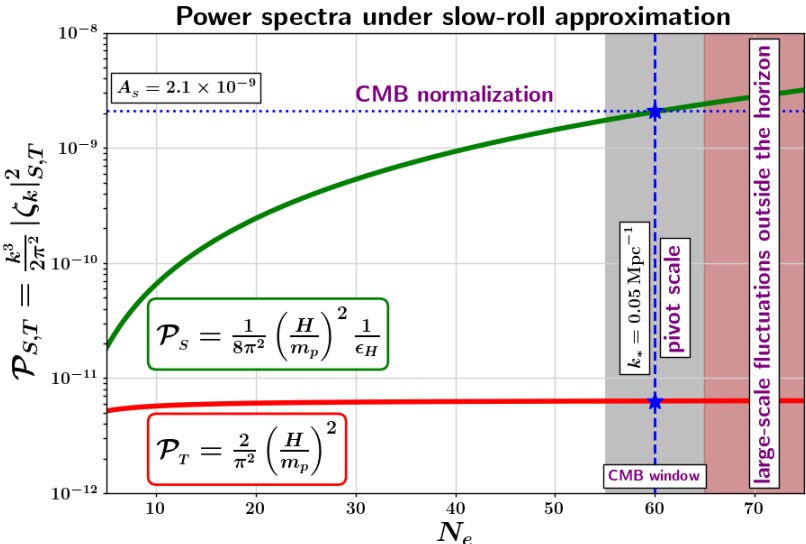

Figure 8: The power spectra of scalar and tensor quantum fluctuations (computed using the slow-roll formulae (27) and (43) respectively) are shown for comoving modes exiting the Hubble radius at different number of e-folds $N_e$ before the end of inflation for Starobinsky potential (58). The CMB window (shown in grey shaded region) corresponds to comoving modes in the range $k_{\mathrm{CMB}} \in [0.0005, 0.5]$ Mpc$^{-1}$ that are being probed by the current CMB missions.

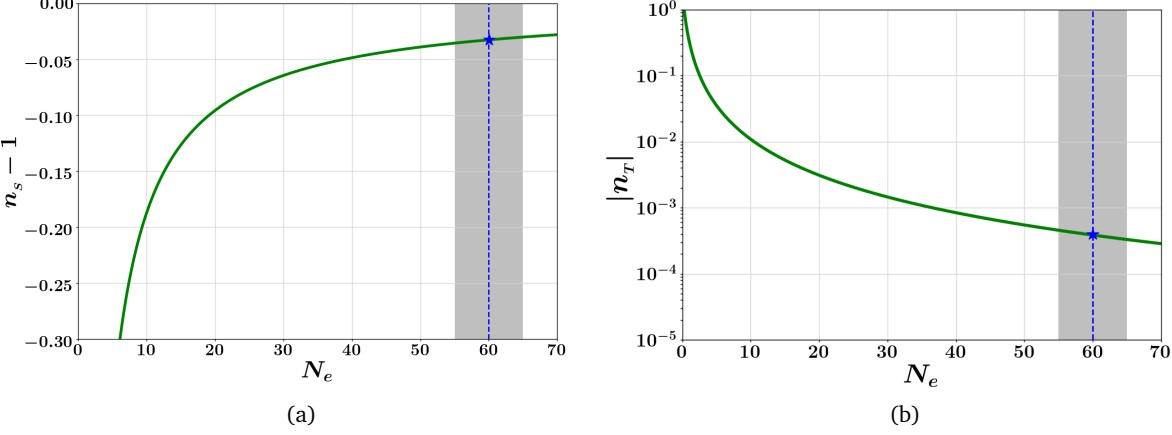

Figure 9: The scalar and tensor spectral indices $n_s$ and $n_T$ are shown as a function of $N_e$ for Starobinsky potential (58) as determined by their slow-roll approximated formulae (44) and (45) respectively. Around the pivot scale, they take the approximate values $n_s \simeq 0.967$ and $n_T \simeq -0.0004$. Note that we have plotted $n_s - 1$ (instead of $n_s$) since it is the correct scalar spectral index. The tensor-to-scalar ratio is given by $r \simeq -8\, n_T$.

## 5  Numerical analysis for quantum fluctuations during inflation

In the previous section we used slow-roll approximated formulae to study the spectra of inflationary fluctuations in terms of background quantities such as $H$, $\epsilon_H$, $\eta_H$. Hence, we only had to simulate the background

dynamics for a given potential in order to plot the relevant inflationary observables. However, if we want to analyze the behaviour of quantum fluctuations more accurately, especially in situations where one or both the slow-roll conditions (15) are violated, we need to numerically solve the Mukhanov-Sasaki Eqn. (21) corresponding to each comoving scale $k$.

For this purpose, we first rewrite the Mukhanov-Sasaki Eqn. (21) in cosmic time as

$$\frac{\mathrm{d}^2 v_k}{\mathrm{d}t^2} + H\frac{\mathrm{d}v_k}{\mathrm{d}t} + \left[\frac{k^2}{a^2} - \frac{1}{a^2}\frac{z''}{z}\right]v_k = 0\,. \tag{80}$$

Note that here $z$ is not the dimensionless Hubble parameter used in our numerical code, rather it is the variable $z = am_p\sqrt{2\epsilon_H}$ in the Mukhanov-Sasaki Eqn. (21). The effective mass term $z''/z$ in (23) can be re-written as

$$\frac{z''}{z} = a^2\left[\frac{5}{2}\frac{\dot{\phi}^2}{m_p^2} + 2\frac{\dot{\phi}\ddot{\phi}}{Hm_p^2} + 2H^2 + \frac{1}{2}\frac{\dot{\phi}^4}{H^2 m_p^4} - V_{,\phi\phi}(\phi)\right]\,. \tag{81}$$

Since $v_k$ is a complex valued function, it is convenient to split it into its real and imaginary parts to study their evolution separately for the numerical analysis. While both will follow the same evolution equation, they will be supplied with different initial conditions in the form of the real and imaginary parts of the Bunch-Davies vacuum (25). Writing the Mukhanov-Sasaki equation for scalar fluctuations in terms of dimensionless variables, we obtain

$$\frac{\mathrm{d}^2 v_k}{\mathrm{d}T^2} + z\frac{\mathrm{d}v_k}{\mathrm{d}T} + \left[\frac{k^2}{A^2} - \frac{5}{2}y^2 + 2\frac{y}{z}\left(3zy + \frac{v_0}{S^2}f_{,x}\right) - 2z^2 - \frac{1}{2}\frac{y^4}{z^2} + \frac{v_0}{S^2}f_{,xx}\right]v_k = 0\,. \tag{82}$$

Our primary goal in this section is to numerically solve Eqn. (82) for the Fourier modes $v_k$ corresponding to each comoving scale $k$ and plot the frozen value of the scalar power spectrum of $\zeta_k$ given by (26) after the mode becomes super-Hubble. We can conveniently relate a comoving scale $k$ to its Hubble-exit epoch by $k = aH$. Since we are only interested in the super-Hubble power spectra, we only need to simulate the system to evolve $v_k$ for a relatively shorter duration of time[14] around the Hubble-exit of scale $k$.

In the following, we discuss the algorithm to solve the Mukhanov-Sasaki Eqn. (82) and determine the scalar power spectrum (26) numerically. We also discuss how to solve the corresponding Eqn. (35) for the tensor power spectrum (at linear order in perturbation theory)[15]. The dimensionless Mukhanov-Sasaki equation for tensor fluctuations is given by

$$\frac{\mathrm{d}^2 h_k}{\mathrm{d}T^2} + z\frac{\mathrm{d}h_k}{\mathrm{d}T} + \left[\frac{k^2}{A^2} + \frac{1}{2}y^2 - 2z^2\right]h_k = 0\,. \tag{83}$$

We explicitly write down the Mukhanov-Sasaki equations for scalar and tensor fluctuations in terms of dimensionless variables (as used in our code) in the following way

---

[14]Keeping in mind the important caveat that the initial time should be sufficiently early enough to impose Bunch-Davies initial conditions, and the final time should be sufficiently late enough for the mode to be frozen outside the Hubble radius. As discussed below, for most of the potentials considered in this work, as well as for most single-field slow-roll violating models [68] relevant for PBH formation in the literature, imposing initial conditions for about 5-6 e-folds before the end of inflation is sufficient.

Nevertheless, in general, especially for slow-roll violating models, one might need to evolve the mode functions for longer duration and our numerical set-up easily allows the user to incorporate a longer evolution duration for the mode functions.

[15]Second-order tensor fluctuations which are induced by first-order scalar fluctuations will be discussed in the revised version of our manuscript.

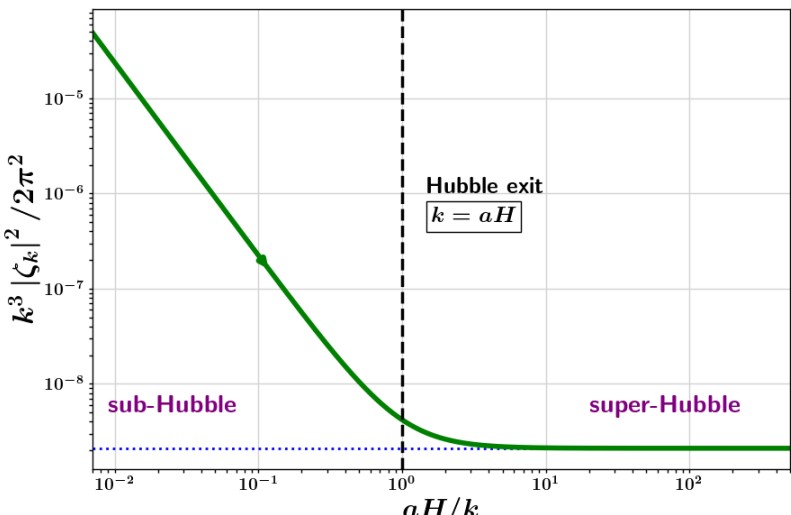

Figure 10: Evolution of scalar power $\frac{k^3}{2\pi^2}|\zeta_k|^2$ is plotted by numerically solving the Mukhanov-Sasaki Eqn. (82) for a mode exiting the Hubble radius at about 60 e-folds before the end of inflation (for Starobinsky potential). At early times when the mode is sub-Hubble, *i.e.* $k \gg aH$, the power decreases as $\mathcal{P}_\zeta \sim (aH)^{-2}$ as expected. After the Hubble-exit, the power freezes to a constant in the super-Hubble regime when $k \ll aH$. We note down its value after the mode-freezing as the super-Hubble scale power corresponding to that mode. Repeating the procedure for a range of scales $k$ yields us the power spectrum of scalar fluctuations. The same numerical analysis can be carried out for tensor fluctuations.

$$v_{k,T} = \frac{\mathrm{d}v_k}{\mathrm{d}T}, \tag{84}$$

$$\frac{\mathrm{d}v_{k,T}}{\mathrm{d}T} = -z\, v_{k,T} - \left[ \frac{k^2}{A^2} - \frac{5}{2}\, y^2 + 2\, \frac{y}{z}\left(3z\, y + \frac{v_0}{S^2}\, f_{,x}\right) - 2z^2 - \frac{1}{2}\, \frac{y^4}{z^2} + \frac{v_0}{S^2}\, f_{,xx} \right] v_k \,; \tag{85}$$

$$h_{k,T} = \frac{\mathrm{d}h_k}{\mathrm{d}T}, \tag{86}$$

$$\frac{\mathrm{d}h_{k,T}}{\mathrm{d}T} = -z\, h_{k,T} - \left( \frac{k^2}{A^2} + \frac{1}{2}\, y^2 - 2z^2 \right) h_k \,. \tag{87}$$

In our numerical set up, we split $v_k$ and $h_k$ into their real and imaginary parts and simulate them separately with appropriate Bunch-Davies initial conditions. We begin with a discussion of numerical simulations of the Mukhanov-Sasaki equations (82) and (83) for a purely slow-roll potential (which we choose to be the Starobinsky potential (58) as usual), before moving forward to discuss the same for a potential with a slow-roll violating feature. This latter case is of the primary focus of our paper. In particular, we will illustrate our numerical scheme for the case of a base slow-roll potential possessing a tiny local bump feature, which was proposed in [44] in the context of PBH formation.

## 5.1 Numerical analysis for slow-roll potentials

1. As the first step, we numerically solve the background dynamics for a given potential, determine the values of all relevant parameters of the potential and the evolution of relevant primary dynamical variables $\{x, y, z, A\}$ as well as the derived quantities such as $\{N_e, \epsilon_H, \eta_H\}$ (as discussed in section 4).

2. We then proceed to identify different comoving scales $k$. This can be done by determining their Hubble-exit epochs in the following way. For example, we plot $aH$ (in log scale) against $N_e$ and identify the value of $aH$ at $N_e = N_*$ to be the CMB pivot scale $k_p$. As mentioned before, we take $N_* = 60$ in all our analysis. Similarly, we associate a corresponding value of $N_e$ to each comoving scale $k$ by the value of $aH$ at its Hubble-exit epoch. This step ensures that we have a one-to-one correspondence between $k$ and $N_e$ in our analysis and we can use them interchangeably.

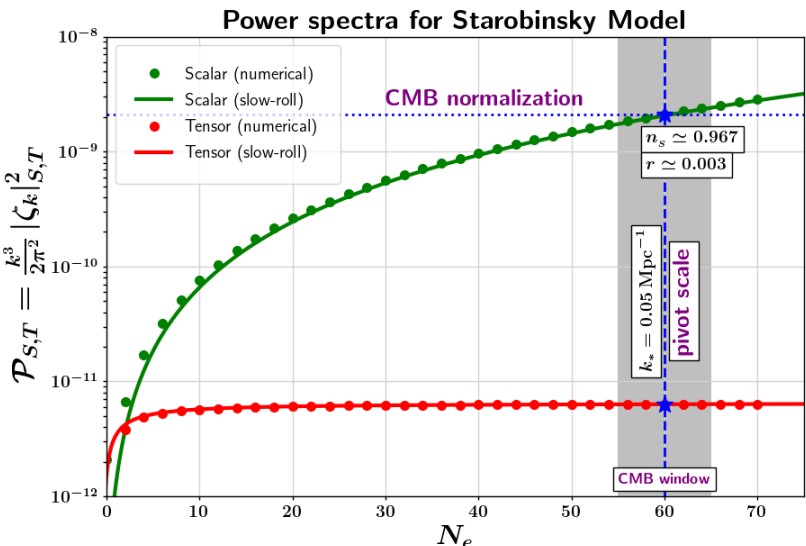

Figure 11: The super-Hubble power spectra of scalar fluctuations (in green colour) and tensor fluctuations (in red colour) are plotted for modes exiting the Hubble radius at different number of e-folds $N_e$ before the end of inflation for Starobinsky potential (58). The solid curves represent power spectra computed under the slow-roll approximation (27), while the dotted curves represent the power computed by numerically solving the Mukhanov-Sasaki Eqn. (82). We conclude that for Starobinsky model, since slow-roll conditions $\epsilon_H, |\eta_H| \ll 1$ are easily satisfied for most part of inflation, the power spectra computed under the slow-roll approximation match quite well with their numerically determined counterparts.

3. We intend to impose Bunch-Davies initial conditions for a given mode $v_k$ at an epoch when it is sub-Hubble. As it tuns out, for most potentials, the Bunch-Davies initial conditions can be safely imposed as long as $k \geq 100 \, aH$. Hence, rather than simulating the Mukhanov-Sasaki equation for each mode (making Hubble-exit at the corresponding value of $N_e$) all through the inflationary history (starting from $\phi_i > \phi_*$), we actually impose the initial conditions from the background solutions for $\{x, y, z, A\}$ at around 5 e-folds before the Hubble-exit of that mode. This step greatly reduces the running-time of the code. We then incorporate the initial value of scale factor $A_i$ at the same epoch, namely $A_i \exp(N_T - N_e - 5)$ and do the same for the initial values of the field $x_i$, and its derivative $y_i$. The initial conditions for the mode functions $v_k$ and their derivatives $\dot{v}_k$ can then be safely taken to be of Bunch-Davies type (however, see the caveat given in footnote 14).

4. We solve the set of cosmological equations with these initial conditions for a period of time $T = T_i \rightarrow T = T_f$ such that the mode becomes super-Hubble and its power $(k^3|\zeta_k|^2/2\pi^2)$ is frozen to a constant value (see Fig. 10), which is typically within 5 e-folds after Hubble-exit in the kind of models we are interested in. We note down this frozen value as the value of the power spectrum of that mode. While we have been discussing about scalar fluctuations mostly, the same can be done for tensor fluctuations which we have incorporated in our code.

5. We then select another mode that leaves the Hubble radius at some epoch $N_e$ and repeat the procedure until we have collected the frozen super-Hubble power spectra of a range of scales that we are interested in (see Fig. 11).

From the above numerical analysis of featureless vanilla potentials which exhibit slow-roll dynamics until close to the end of inflation, we observe that the power spectra of scalar and tensor fluctuations are nearly scale-invariant (with small red-tilt) and their behaviour (as obtained from numerically solving the Mukhanov-Sasaki equation) matches quite well with the analytical predictions under the slow-roll approximations (see Fig. 11).

However, for potentials exhibiting a small-scale feature at intermediate field values $\phi < \phi_*$, there might exist a short period of slow-roll violating phase before the end of inflation during which slow-roll approximations break down. In particular, as we will see, while the first slow-roll parameter remains small $\epsilon_H \ll 1$, the second slow-roll parameter might become $\eta_H \sim \mathcal{O}(1)$. Hence, a numerical analysis of the Mukhanov-Sasaki equation is desired in order to determine the scalar power spectrum more accurately. This will be the main focus of discussion in the next subsection.

## 5.2 Numerical analysis for potentials with a local bump/dip feature

In order to facilitate PBH formation, we need a large amplification of scalar power spectrum at smaller scales during inflation. This can be achieved by introducing a small-scale feature in the potential which leads to a transient period of slow-roll violating phase (including a short almost-USR phase). Adequate amplification in the super-Hubble scalar power spectrum results in a large density contrast in the post-inflationary universe (upon the Hubble-entry of the corresponding modes) which in turn can collapse to form PBHs. Usually, such a PBH feature in the potential leads to an increase in the value of the second slow-roll parameter $\eta_H$ from near-zero to a positive value of $\eta_H \sim \mathcal{O}(1)$.

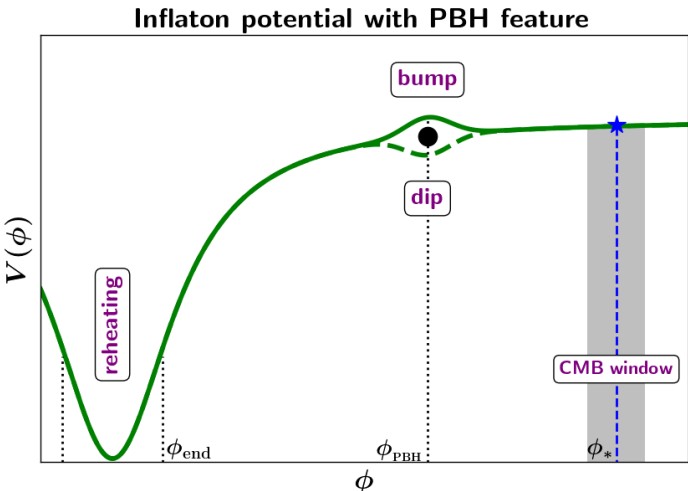

Figure 12: This is a schematic plot of an asymptotically-flat inflationary potential with a tiny bump/dip feature (88) near intermediate field values $\phi \simeq \phi_{\mathrm{PBH}}$ that leads to an enhancement of the scalar power spectrum. The full potential asymptotes to the base (slow-roll) potential near the CMB window $\phi \simeq \phi_*$, thus satisfying observational constraints on large cosmological scales. Slow-roll is violated around the feature, whose position $\phi_{\mathrm{PBH}}$ dictates the range of moving scales $k$ that receive amplification of power (which accordingly determines the mass and abundance of formed PBHs). Note that the feature has been greatly exaggerated for illustration purpose. In most realistic models, both the height and the width of the feature are too small to be seen (without zooming-in considerably).

A number of models with different types of features have been proposed in the recent literature (as mentioned before) that facilitate the amplification of scalar power spectrum at small scales. The most common amongst them is an inflection point-like feature. However, we choose the model proposed in [44] in which the base inflaton potential $V_b(\phi)$ possesses a tiny local bump or dip $\pm \varepsilon(\phi, \phi_0)$ at an intermediate field value $\phi_0$ of the form

$$V(\phi) = V_b(\phi) \left[ 1 \pm \varepsilon(\phi, \phi_0) \right], \tag{88}$$

where we assume $V_b(\phi)$ to be a symmetric or an anti-symmetric asymptotically-flat potential in order to satisfy CMB constraints at large cosmological scales. Such a potential has been schematically illustrated in Fig. 12. To be specific, in this paper we choose the base potential to be the D-brane KKLT potential [80–83] with a tiny Gaussian bump of the form [44]

$$V(\phi) = V_0 \frac{\phi^2}{m^2 + \phi^2} \left[ 1 + A \exp\left( -\frac{1}{2} \frac{(\phi - \phi_0)^2}{\sigma^2} \right) \right], \tag{89}$$

where $m$ is a mass scale in the KKLT model, while $A$ and $\sigma$ represent the height and the width of the tiny bump respectively. We use this particular model to demonstrate our numerical framework because of its simplicity and efficiency. However, one can choose any model of their interest. Values of all the parameters appearing in (89), which we use in our numerical analysis, have been explicitly shown in Fig. 13.

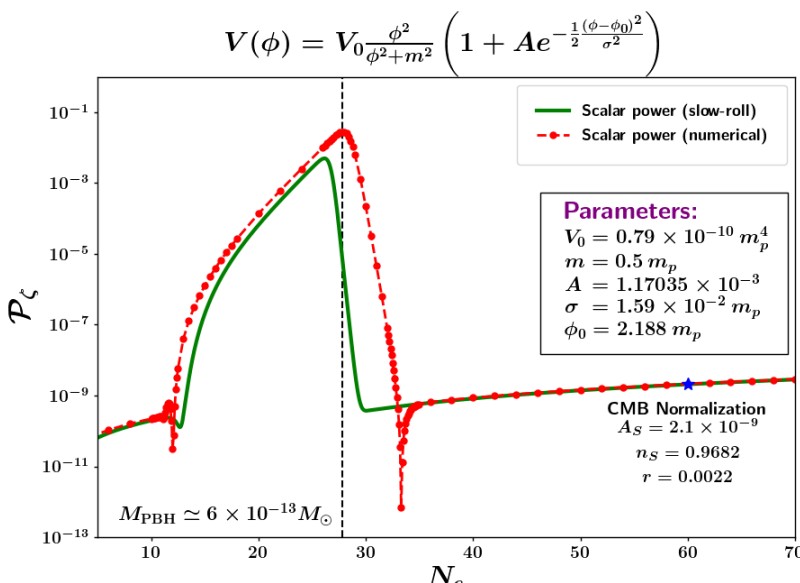

Figure 13: The super-Hubble power spectra of scalar fluctuations are plotted for modes exiting the Hubble radius at different number of e-folds $N_e$ before the end of inflation for KKLT potential with a tiny bump (89). The solid green curve represents the power spectrum computed under the slow-roll approximation (27), while the dotted red curve represents the power computed by numerically solving the Mukhanov-Sasaki Eqn. (82).

The height and the width of the (bump/dip) feature required to facilitate adequate amount of power amplification are quite small, and hence the feature is tiny and local (in contrast to inflection point-like features). This ensures that the feature does not significantly affect the CMB observables. However, since we gain a lot of extra e-folds of expansion $\Delta N_e \simeq 15$ with little change in the field value (as shown in Fig. 14) when the inflaton crosses the feature, the CMB pivot scale gets shifted towards smaller values as compared to the same for the base potential.

Since slow-roll is violated in these models (as shown in Fig. 15), we need to solve the Mukhanov-Sasaki equation numerically in order to accurately compute the scalar power spectrum. This has been explicitly

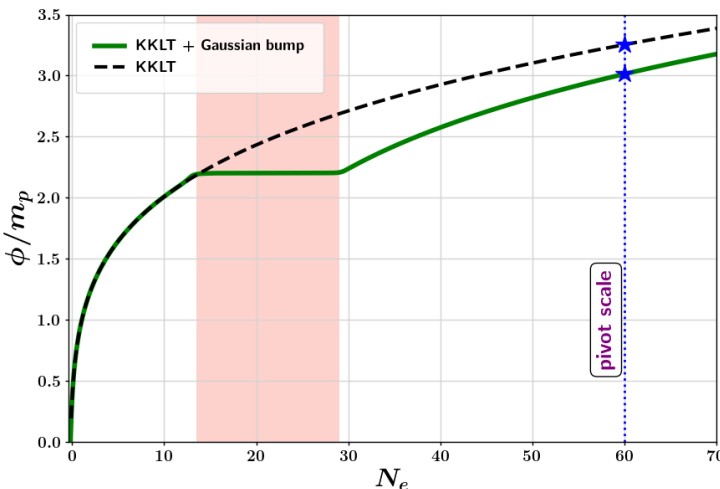

Figure 14: Evolution of the field value $\phi$ for the KKLT potential with a tiny bump (89) is shown in solid green curve as a function of number of e-folds $N_e$ before the end of inflation. We note that at CMB scales, $\phi$ is much smaller than the corresponding value in the base model (KKLT) (shown in dashed black curve). At intermediate scales when the inflaton evolves across the bump feature in the potential, we gain a lot of extra e-folds of expansion $\Delta N_e \simeq 15$ with little change in the field value. After crossing the feature, evolution of $\phi$ mimics its corresponding value in the base model.

demonstrated in Fig. 13. Note that the inflationary dynamics in such models contains a number of phases that include an early slow-roll phase **SR-I** near the CMB window, a transition **T-I** from the early **SR-I** to the subsequent almost ultra slow-roll phase **USR** and a transition **T-II** back to the next slow-roll phase **SR-II** after passing through an intermediate constant-roll phase **CR** (shown in Fig. 15).

The effective mass term $z''/z$ in the Mukhanov-Sasaki Eqn. (21), which primarily governs the dynamics of scalar fluctuations, has been shown in Fig. 16, and the resultant Hubble-exit behaviour of different modes is described in Fig. 17. As the inflaton approaches the PBH feature, $\eta_H$ starts to increase rapidly. This is accompanied by an initial sharp dip in the effective mass term $z''/z$, which then increases to a higher plateau as $\eta_H$ approaches its maximum in the USR type phase. The modes that leave the Hubble radius slightly before the transition already start receiving power amplification on super-Hubble scales as shown by the orange colour curve in Fig. 17.

It is worth noting that the power spectrum exhibits a dip which corresponds to very narrow range of scales $k \simeq k_{\rm dip}$ that leave the Hubble radius a few e-folds before the USR phase (shown by the blue color curve in Fig. 17). The maximum rate of growth observed in this model is consistent with the steepest growth bound discussed in [76]. Near the USR phase when $\eta_H \gtrsim 3$, $z''/z$ saturates to a constant value. Modes leaving the Hubble radius around this USR epoch receive maximal amplification in their super-Hubble power spectrum.

As the field crosses the maximum of the bump feature, $\eta_H$ decreases to a constant negative value. It stays in this constant-roll phase until the inflaton meets the base potential eventually and approaches the final slow-roll phase in its dynamics before the end of inflation.

The dynamics of scalar fluctuations in the aforementioned phases are quite rich, and interesting. However, since the main aim of this paper is to illustrate how to use our numerical code with an example, we do not discuss these phases and their impact on the power spectrum (some of which have been explicitly shown in Figs. 16, 17, 18), and refer the interested readers to [68] for more detail.

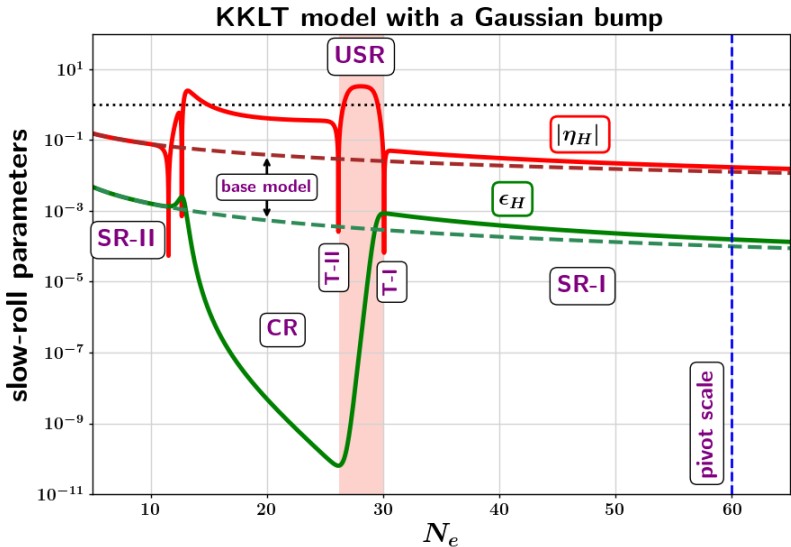

Figure 15: Evolution of the slow-roll parameters $\epsilon_H$ and $\eta_H$ is shown in solid green and solid red curves respectively for the KKLT potential with a tiny bump (89). Both $\epsilon_H$ and $\eta_H$ are close to their corresponding values for the base KKLT potential (shown in dashed curves) at early times near the CMB window. At $N_e \simeq 30$, the value of $\epsilon_H$ starts decreasing rapidly leading to an increase in $\eta_H$ from $|\eta_H| \ll 1$ to a higher and positive value $\eta_H \simeq +3.3$ (almost USR phase). Thereafter, the inflaton enters a phase of constant-roll inflation (where $\eta_H \simeq -0.37$) before returning to the final slow-roll phase.

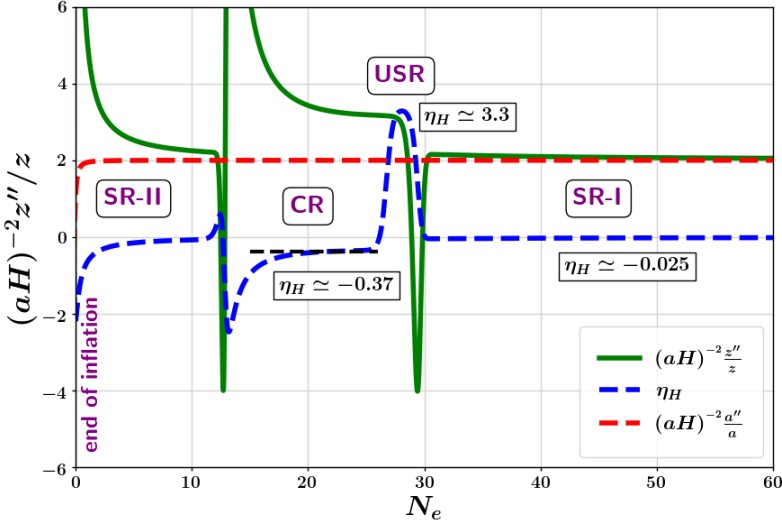

Figure 16: Evolution of the effective mass term in the Mukhanov-Sasaki equation is shown here in solid green curve for the KKLT potential with a tiny bump (89). Important transient phases of the scalar field dynamics are also highlighted following the behaviour of the second slow-roll parameter $\eta_H$ (plotted in dashed blue curve). There is a sharp dip in the effective mass term when the field transitions from the first slow-roll phase (SR-I) to an almost ultra slow-roll phase (USR). The inflaton later makes a transition to a phase of constant-roll inflation with $\eta_H \simeq -0.37$, before reaching a final slow-roll phase until the end of inflation.

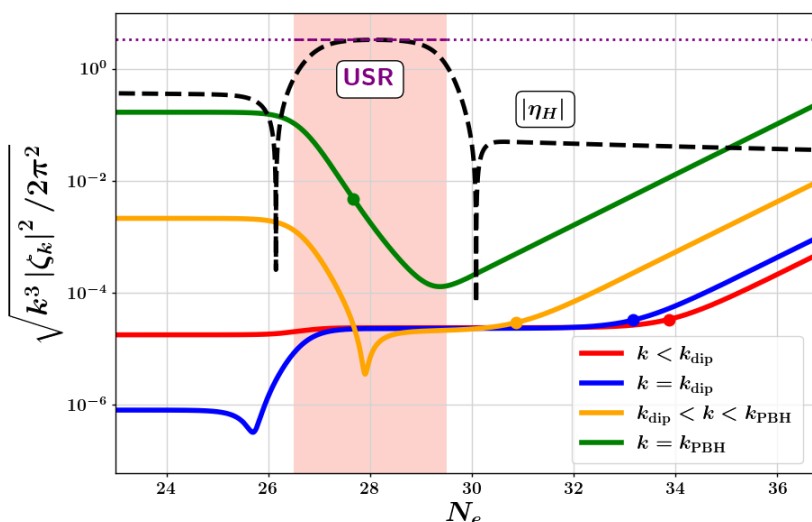

Figure 17: This figure demonstrates the horizon exit behaviour of different modes *i.e* the evolution of $\sqrt{\mathcal{P}_\zeta}$ for different modes as they cross the Hubble radius in KKLT model with a Gaussian bump (89). The dot on each curve corresponds to its Hubble-exit epoch. The sharp dip in the power spectrum (Fig. 13) corresponds to the mode $k_{\text{dip}}$ (plotted in blue color) that exits the Hubble radius a few e-folds before the commencement of the USR phase. The mode $k_{\text{PBH}}$ which exits the Hubble radius during the USR phase receives a maximal amplification of power (plotted in green color).

# 6  Comparison of the numerical efficiency of different variables

In the aforementioned numerical analysis, we utilized the Mukhanov-Sasaki variable $v_k$, with its evolution Eq. (21), in order to simulate the scalar fluctuations $\zeta_k$ at linear order in perturbation theory. However, the power spectrum can be determined by using a number of other variables proposed in the literature. In this section, we use three different variables to compute the power spectrum for the KKLT potential with a tiny Gaussian bump, as given in Eq. (89), with parameters as displayed in Fig. 13. The first one is the curvature perturbation $\zeta_k = v_k/z$ whose evolution is described by

$$\zeta_k'' + 2\left(\frac{z'}{z}\right)\zeta_k' + k^2\zeta_k = 0\,. \tag{90}$$

The authors of Ref. [84] have suggested[16] a different variable $g_k = v_k e^{ik\tau}/z$ which is supposed to make the numerical simulation much more stable by removing the early time oscillations by using the phase term $e^{ik\tau}$. The corresponding evolution equation for $g_k$ is given by

$$\ddot{g}_k + \left(H + 2\frac{\dot{z}}{z} - \frac{2ik}{a}\right)\dot{g}_k - \left(\frac{2ik}{a}\right)\frac{\dot{z}}{z}g_k = 0\,. \tag{91}$$

Similarly, the authors of Ref. [85] have suggested the variable $q_k = a^{1/2}v_k$ which supposedly makes the computation of the density perturbations during preheating much easier to study. Its evolution equation is given by

$$\ddot{q}_k + \left[\frac{k^2}{a^2} + \frac{\mathrm{d}^2 V(\phi)}{\mathrm{d}\phi^2} + 3\dot\phi^2 - \frac{\dot\phi^4}{2H^2} + \frac{3}{4}\left(\frac{\dot\phi^2}{2} - V(\phi)\right) + 2\frac{\dot\phi}{H}\frac{\mathrm{d}V(\phi)}{\mathrm{d}\phi}\right]q_k = 0\,. \tag{92}$$

---

[16]We are thankful to Christian Byrnes for bringing this to our attention.

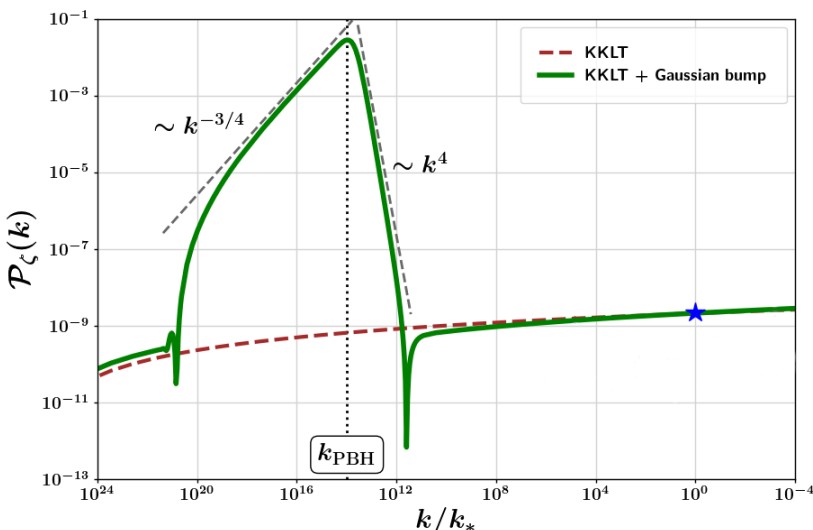

Figure 18: The super-Hubble power spectrum of scalar fluctuations obtained by numerically solving the Mukhanov-Sasaki Eqn. (82) is plotted here for the KKLT potential with a tiny bump (89). At large scales (around the CMB window), the amplitude of the power spectrum roughly matches that of the base KKLT potential, although its spectral index is shifted due to a gain in the number of e-folds because of the presence of the bump, see Ref. [44] for a detailed discussion on this. The power spectrum (after exhibiting a sharp dip) receives a large amplification at intermediate scales $k \sim k_{\text{PBH}}$ around the USR phase. The maximum rate of growth observed in this model is consistent with the steepest growth bound discussed in [76]. After reaching the peak, the power then decreases at a steady rate during the constant-roll phase before finally asymptoting towards its base slow-roll value towards the end of inflation.

We compute the power spectrum of $\zeta_k$ numerically by using each of the aforementioned variables and compare the time it takes for our code run in each case. Note that the power spectrum of $g_k$ is the same as that of $\zeta_k$ since they only differ by a factor $e^{ik\tau}$, namely,

$$\mathcal{P}_\zeta(k) = \frac{k^3}{2\pi^2}|\zeta_k|^2 = \frac{k^3}{2\pi^2}|g_k|^2 = \frac{k^3}{2\pi^2}\frac{|v_k|^2}{2a^2\epsilon_H} = \frac{k^3}{2\pi^2}\frac{|q_k|^2}{2a^3\epsilon_H}.$$ (93)

The Bunch-Davies initial conditions for the the different variables when a mode is sub-Hubble are given by:

$$v_k: \qquad v_i = \frac{1}{\sqrt{2k}} \qquad\qquad \dot{v}_i = -\frac{1}{a_i}\frac{ik}{\sqrt{2k}}$$ (94)

$$\zeta_k: \qquad \zeta_i = \frac{1}{z}\frac{1}{\sqrt{2k}} \qquad\qquad \dot{\zeta}_i = \frac{1}{\sqrt{2k}}\left[-\frac{1}{z}\frac{\dot{z}}{z} - \frac{ik}{az}\right]$$ (95)

$$g_k: \qquad g_i = \frac{1}{z}\frac{1}{\sqrt{2k}} \qquad\qquad \dot{g}_i = -\frac{1}{z}\frac{\dot{z}}{z}\frac{1}{\sqrt{2k}}$$ (96)

$$q_k: \qquad q_i = \sqrt{\frac{a}{2k}} \qquad\qquad \dot{q}_i = \frac{1}{\sqrt{2k}}\left[\sqrt{a}H - \frac{ik}{\sqrt{a}}\right]$$ (97)

We numerically solve the evolution equations of the four variables and note down the run-time of our code for different Fourier modes in each case. Tab. 1 and Fig. 19 show the variation in the percentage of

fractional differences in the compilation times for the variables $\zeta_k$, $g_k$ and $q_k$, relative to the Mukhanov-Sasaki variable $v_k$ for the evolution of two distinct modes, namely, $k_{\mathrm{PBH}}$ and $k_{\mathrm{CMB}}$, since they are the primary modes of our interest.

| Variable | $\frac{\Delta T}{T}\big|_{\mathrm{CMB}}$ | $\frac{\Delta T}{T}\big|_{\mathrm{PBH}}$ |
|---|---|---|
| $v_k$ | - | - |
| $\zeta_k$ | 71% | 93% |
| $g_k$ | ! | 70% |
| $q_k$ | 55% | 64% |

Table 1: The percentage of fractional differences in the compilation times for different variables, such as $\zeta_k$, $g_k$ and $q_k$, relative to the Mukhanov-Sasaki variable $v_k$ have been tabulated here.

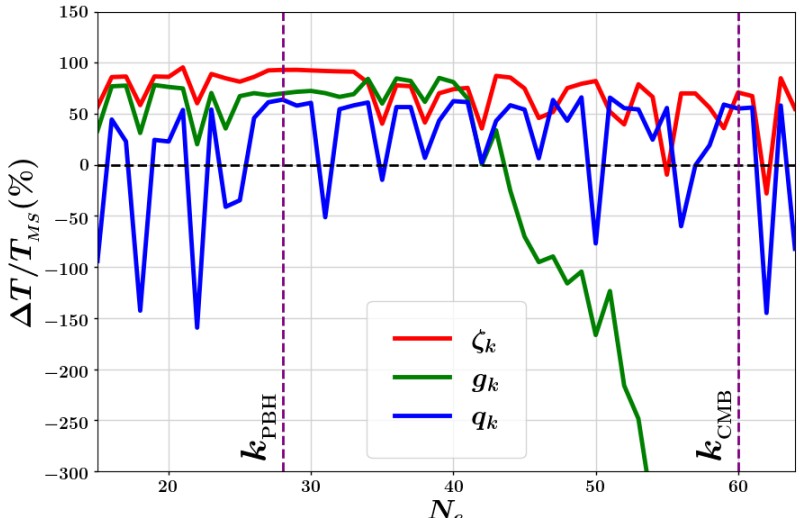

Figure 19: The percentage of fractional differences in the compilation times for different variables, such as $\zeta_k$ (in red colour), $g_k$ (green colour) and $q_k$ (blue colour), relative to the Mukhanov-Sasaki variable $v_k$ is shown in this figure. Number of e-folds corresponding to the Hubble-exit epochs of the CMB pivot scale and the PBH scale have been shown by the purple colour dashed lines.

The curvature perturbation $\zeta_k$ gets computed the fastest, with the code running nearly twice as fast as that for the Mukhanov-Sasaki variable for modes around $k_{\mathrm{PBH}}$ as well as around $k_{\mathrm{CMB}}$. The Räsänen-Tomberg variable ($g_k$) is also substantially faster for modes around $k_{\mathrm{PBH}}$, but we find that it is much slower for modes that exit the Hubble radius much earlier than $k_{\mathrm{PBH}}$. Around $k_{\mathrm{CMB}}$ the compilation takes a drastically long time and it can't be practically compared to that of the Mukhanov-Sasaki variable. This can be potentially attributed to the factor $e^{ik\tau}$ being blowing up rapidly, and hence, requiring computation of very large numbers. The Finelli-Brandenberger variable ($q_K$) is relatively erratic and runs faster for some modes and slower for others. Our results indicate that $\zeta_k$ is the best-suited variable for carrying out numerical simulations during inflation. Moreover, the Finelli-Brandenberger variable ($q_k$) is not very suitable to compute inflationary dynamics, and it is also not expected to be so, as it was introduced in the context of post-inflationary oscillations.

Upon repeating the comparison between the aforementioned variables for a different set of parameters,

we find that although the precise time differences for the variables may be vary slightly, their behaviour is found to be concordant especially around the modes $k_{\mathrm{PBH}}$ and $k_{\mathrm{CMB}}$. It should be noted that the conclusions drawn in this section can be dependent upon the specifications of the computer processor. The absolute compilation times for all the variables discussed above were only about a few seconds. It is possible that, with faster systems, the difference in the compilation times is small enough that the trends observed differ vastly. However, care has been taken to minimize any such confounding due to other running processes, unstable power source, *etc*, and the codes have been tested on different systems. The numbers reported are unique to a generic personal computer, which in our case is a **Lenovo IdeaPad 310** with **8GB DDR4 RAM** running **Ubuntu 20.04** OS with a **Python 3.10** compiler.

## 7  Future extension of our numerical framework

In preceding sections, we described the relevant cosmological equations governing the inflationary dynamics in terms of dimensionless variables. We also introduced our numerical code (written in terms of cosmic time) that can easily simulate the inflationary dynamics both at the background level in section 4 and at linear order in perturbation theory in section 5. However, with minimal to moderate extension, our numerical code can be used to simulate a number of different scenarios associated with scalar field dynamics both during inflation as well as in the post-inflationary universe. In the following we discuss some important future extensions of our code that we intend to include in the near future.

1. As stressed in section 4, the present version of our numerical code, although quite fast and neat, contains segments that require the user to carry out a number of tasks manually. While we believe that the present version will definitely help a user (who is relatively new to the field) to understand the inflationary dynamics much better, we are already developing an automated version of this code that is much more compact and requires substantially less manual involvement of the user. We also plan to make the code even faster. We will present the updated version of our code it in the same `GitHub link` [17] in the near future.

2. In section 5, we briefly discussed how to solve the evolution equation for the tensor fluctuations at linear order in perturbation theory. However, first-order scalar fluctuations induce tensor fluctuations at second order in perturbation theory which might be significant when slow-roll is violated, especially in the scenario where scalar power spectrum is largely amplified in order to source PBH formation. We are planning to incorporate the computation of such scalar-induced Gravitational Waves in the updated version of our code.

3. It is easy to extend our numerical analysis to incorporate the dynamics of more than one scalar fields during inflation, at least in the background level. In the future, we are going to provide an updated numerical code to study both the background dynamics as well as quantum fluctuations in two-field inflationary dynamics (where the second field might also source inflation, or act as a spectator field).

4. Additionally, our code can be extended to simulate the post-inflationary dynamics of the inflaton field as well as to study parametric resonance by simulating the evolution of different Fourier modes of the inflaton fluctuations. During the post-inflationary oscillations, it is usually advisable to make a change in the Mukhanov-Sasaki variable of the form $\tilde{v}_k = a^{1/2} v_k$ as suggested in [86, 87]. The code can also be extended to study the dynamics of scalar field dark matter and quintessence by suitably redefining the dimensionless variables as per the energy scale of the dynamics.

---

[17] https://github.com/bhattsiddharth/NumDynInflation

# 8  Discussion

In this paper, we introduced our numerical approach to simulate the cosmological equations in order to study the inflationary dynamics both at the background level and at linear order in perturbation theory. We provided the link to our open-source `GitHub` repository where we have supplied a `Python`-based simple numerical code to simulate inflationary dynamics in terms of cosmic time $t$. We explicitly demonstrated how to use the code to study the inflationary background dynamics in section 4 that includes plotting the phase-space portrait of inflation as well as to characterise quantum fluctuations during inflation using the simulations of the background dynamics.

Section 5 was dedicated to study quantum fluctuations during inflation (without using slow-roll approximated expressions) by numerically solving the mode function equations of scalar and tensor fluctuations. For a featureless slow-roll inflaton potential, the difference between the results obtained numerically and those obtained under slow-roll approximations were negligible until close to the end of inflation, as expected. We used the Starobinsky potential (58) as an example to illustrate our analysis. Our primary focus was the numerical evaluation of the scalar power spectrum $\mathcal{P}_\zeta(k)$ for potentials that exhibit a slow-roll violating feature. In particular, we used the example of an asymptotically-flat base inflationary potential that possesses a tiny local bump feature (88) to illustrate our numerical scheme in section 5.2. By suitably choosing the parameters of the potential (89), one can achieve a large enough amplification of the scalar power spectrum in order to facilitate the formation of PBHs in the post-inflationary epoch.

In our numerical analysis for the case of potentials with a slow-roll violating feature, we explicitly chose the parameters in order to amplify the small-scale scalar power by a factor of $\sim 10^7$ with respect to the corresponding power at large cosmological scales in order to facilitate the formation of PBHs, as per the standard practice. However, it is important to stress that a significant growth in the scalar power spectrum at small-scales might engender the dynamics to enter into non-perturbative regime. For example, a careful computation of loop corrections to the two-point scalar fluctuations demonstrates [88, 89] that the contribution from 1-loop effects might become comparable to the tree-level computation ( carried out in this work) if $\mathcal{P}_\zeta(k) \sim 10^{-2}$, indicating a breakdown of perturbation theory. This is a topic of intense debate at present and we direct the interested readers to Refs. [90–107].

Moreover, the mechanism of PBH formation in the context of single field models of inflation involves additional intricacies that demand for a non-perturbative analysis of primordial fluctuations. A sharp drop in the classical drift speed of the inflaton due to the presence of the PBH-producing feature often trigger the system to enter into a phase where stochastic quantum diffusion effects become important. Furthermore, since PBHs form from rare extreme peaks, their abundance is sensitive to the tail of the probability distribution function (PDF) $P[\zeta]$ of the primordial fluctuations. Hence the standard perturbative computations based on the two-point correlation function or the power-spectrum lead to an inaccurate estimation of the PBH mass fraction. Therefore it is important to determine the full primordial PDF, which can be computed non-perturbatively using the framework of *stochastic inflation*, see Refs. [108–133], which often predicts a non-Gaussian (exponential) tail [114, 118]. Note that the tail behaviour of the primordial fluctuations can also be computed by using semi-classical techniques discussed in [134]. Computation of the tail of the primordial PDF is an active topic of research [118, 134–139] at present, which is beyond the scope of our perturbative analysis presented in this work. However, note that our numerical framework was recently used in Ref. [74] to study the dynamics of noise-matrix elements in stochastic inflation beyond slow roll.

In Sec. 6, we compared the time taken by our numerical computation using different variables proposed in the literature to simulate the linear perturbations during inflation in scenarios where slow roll is violated. We find that although the curvature perturbation $\zeta_k$ shows better compilation time, the Mukhanov-Sasaki variable $v_k$ (which we have used in our analysis) is a good enough variable for all practical purposes.

The numerical framework thus developed, and accompanying notes are aimed at providing a concise but comprehensive `Python` pipeline to solve problems in single-field inflation. Our work introduces a simple and user-friendly numerical framework, which is particularly more suitable for students and researchers who are at a relatively new into carrying out such computations. Several similar, even more sophisticated, frameworks relevant to inflation have been suggested in the literature. For example, Ref. [140] discusses a

python package called *PyTransport*, while Ref. [141] discusses a C++ based platform called *CppTransport* for the numerical computation of inflationary correlators. Similarly, authors of Refs. [142, 143] have developed the Bispectra and non-Gaussianity Operator (BINGO [18]), a set of codes in Fortran and Python that solve inflationary dynamics including the primordial non-Gaussianity parameter $f_{NL}$. Their code is publicly available on GitHub, along with an accompanying Python notebook[19].

Furthermore, it is important to note that our work uses cosmic time $t$ (along with a suitable mass scale to generate a dimensionless variable) as the time variable. In the literature, most frameworks incorporate number of e-folds $N$ as the time variable since it is arguably more intuitive in describing the evolution during inflation. On the other hand, the usage of cosmic time $t$ is quite convenient under certain circumstances. Firstly, for quasi dS (near-exponential) inflation, since $H$ is almost constant, $N \simeq Ht$, therefore both $N$ and $t$ are equally intuitive, especially for vanilla slow-roll models. Additionally, cosmic time is particularly more suitable for describing the post-inflationary oscillations, and hence it can also be used conveniently to simulate the dynamics scalar-field dark matter. Moreover, our framework also describes the study of initial conditions for inflation, where the initial phase-space trajectories may correspond to pre-inflationary (decelerating) epoch. Nevertheless, given a vast majority of numerical frameworks on inflation do utilize $N$, our work provides an alternative time variable. However, it is worth pointing out that computing higher-order correlators as well as quantities at a higher-order in perturbation theory (loop corrections to power spectra), the cosmic time variable might present challenges. Similarly, it may be important to investigate the efficiency of cosmic time as a time variable while comparing models against cosmological data. We plan to return to these issues in the near future.

Before concluding, let us mention that we also stressed upon various important future extensions of our numerical scheme in section 7 that will result in making our code more efficient, and enable us to simulate the scalar field dynamics in a number of interesting scenarios. This includes updating our code to make it more compact and automated, as well as extending it to study the spectrum of scalar-induced gravitational waves, inflaton dynamics in the post-inflationary epoch, multi-field inflationary dynamics, and even scalar field models of dark matter and dark energy. We will incorporate most of these additional features into the numerical framework in the near future. In the meantime, we welcome constructive comments and suggestions as well as queries from interested readers which will help us in improving the quality of our numerical work. We hope our numerical framework will be useful to the fellow researchers in this field.

# 9 Acknowledgements

SSM thanks Satadru Bag, Shabbir Shaikh, and Varun Sahni for crucial inputs during the early stages of development of this code. SSM also thanks Ed Copeland for helpful discussions. The authors are grateful to Parth Bhargava and Sanket Dave for stimulating discussions on various topics related to numerical dynamics discussed in this paper. SSM is supported by a STFC Consolidated Grant [No. ST/T000732/1]. SSB was supported by the INSPIRE scholarship of the Department of Science and Technology (DST), Govt. of India during his Master's thesis work during which a significant portion of this work was carried out. For the purpose of open access, the authors have applied a CC BY public copyright license to any Author Accepted Manuscript version arising.

**Data Availability Statement:** This work is entirely theoretical and has no associated data. All relevant numerical codes can be found in the GitHub account. We have also compiled an IPython notebook that guides the user through the writing and execution of the codes.

---

[18]https://github.com/dkhaz/bingo

[19]https://gitlab.com/dhirajhazra/simple-codes-in-cosmology/-/blob/master/Inflation-Primordial-Perturbation.ipynb

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
