# Peer review of "Numerical simulations of inflationary dynamics: slow roll and beyond"

_SciPost Physics Codebases, doi:SciPost Phys. Codebases 42 (2024) , SciPost Phys. Codebases 42-r1.0 (2024)_

## Round 1 · Referee Report · Anonymous (Referee 1) · 2024-5-11

Strengths

  1. The manuscript is written clearly.
  2. It deals with a topic that is important in the context of primordial cosmology, viz. the dynamics of inflaton and generation of perturbations during inflation.

Weaknesses

  1. The numerical computation of the inflationary power spectrum is a well understood and often implemented technique and the manuscript does not provide any new insights.
  2. There are other publicly available codes, which are likely to be more efficient.
  3. As I have described in my report, I am not convinced that the code developed is efficient, and it will possibly work more efficiently if e-folds is used as the independent variable.
  4. Lastly, in models involving strong departures from slow roll inflation, it would be prudent to evaluate the inflationary spectra close to or at the end of inflation.

Report

In this manuscript, the authors construct a Python code to calculate the scalar and tensor power spectra generated during inflation. The authors consider the simplest of scenarios involving inflation driven by a single canonical scalar field. Apart from the effects that arise on scales relevant for the cosmic microwave background (say, 10^{-4} < k < 1 Mpc^{-1}), the aim is to also take into account effects that occur on small scales (say, k > 10^4 Mpc^{-1}), such as those that lead to the production of significant number of primordial black holes.

The manuscript begins with a broad discussion on inflation (in Sec. 2) before it goes on to analytically derive the standard results for the scalar and tensor power spectra in slow roll inflation (in Sec. 3). Thereafter, it describes the typical scenario involving an epoch of ultra slow roll inflation that leads to enhanced power on small scales and to an increased production of primordial black holes (in Sec. 3.2). Then (in Secs. 4 and 5), the authors discuss the primary topic of the manuscript, viz. the numerical computation of the inflationary background and the scalar and tensor power spectra. If I understand it correctly (see point 4 on page 21), they use a dimensionless version of cosmic time as the independent variable to integrate the equations governing the background and the perturbations [as described in Secs. 4 and 5, see Eqs. (63)-(72) and Eqs. (80)-(87)]. They also explore (in Sec. 6) the evolution of different dependent variables (characterizing the perturbations) to understand if the evolution can be more accurate and/or efficient.

Requested changes

The manuscript is written clearly. But, I have the following comments about the manuscript:

  1. At the outset, I should say that the numerical computation of the inflationary power spectrum is a well understood and often implemented technique and the manuscript does not provide any new insights.

  2. Secondly, it is also well known that working with cosmic time as the independent variable is inefficient and it is more effective to work with e-folds as the independent variable. If the authors wish to develop codes to calculate more involved quantities (such as loop corrections to power spectra, as they indicate in the concluding section) or compare with the cosmological data, then efficiency will become an important issue.

  3. I am surprised that the authors need to choose by hand an initial value of A (see their comment 1 on page 14). I would have thought that the initial value of the scale factor (see Eq. (67) which relates A to the scale factor a) is determined by the condition that the pivot scale, say, k_\ast, leaves the Hubble radius at N_\ast number of e-folds before the end of inflation.

  4. In point 3 on page 21, the authors say that they impose the initial conditions on the perturbations 5 e-folds before the mode leaves the Hubble radius. And, in point 4, they say that they evaluate the spectrum at 5 e-folds after the mode leaves the Hubble radius. While it is largely fine to impose the initial conditions 5 e-folds before Hubble exit, there can be models/situations wherein this may not be adequate (the axion monodromy model comes to mind) and one may have to impose the initial conditions at earlier times. More importantly, if there arise deviations from slow roll --- specifically, strong departures involving a phase of ultra slow roll inflation --- to avoid the transient effects, the power spectrum needs to be evaluated either close to or, preferably, at the end of inflation. It is not clear if the authors do so in the model with a bump that they consider in Sec. 5.2.

I would urge the authors to attend to the above points.

Recommendation

Ask for major revision

  • validity: high
  • significance: ok
  • originality: low
  • clarity: good
  • formatting: good
  • grammar: good

Author:  Swagat Saurav Mishra  on 2024-09-15  [id 4784]

(in reply to Report 1 on 2024-05-11)

Dear Referee, We are thankful to you for your critical and insightful comments. We have attempted to address them in the revised version of our manuscript. Your suggestions have greatly improved the quality of our paper. Specifically, we have addressed the following -

1) We agree with this comment, which is also similar to a comment made by the second referee (Prof. David Wands). Our response is:

We completely agree with the referee(s) that there exist more sophisticated numerical codes in the literature. We have added a discussion on this in the discussion section of our revised manuscript and included two new references to PyTransport and CppTransport frameworks.

However, we would like to stress that our framework is very simple and user-friendly. It is quite fast and works very well for computing two-point functions. We are keen on extending the framework to compute higher-point functions to study non-Gaussianity and we will be happy to report on the chanellenges and results from our analysis in the future.

2) We have added a long paragraph in the discussion section commenting on the usage of cosmic time 't' as the time variable and the potential advantages and disadvantages over the usage of number of e-folds 'N'.

3) We would like to stress that, on page 14, the comment 1 regarding the initial value of A concerns only the background evolution. This does not affect any observable quantity. When we solve the Mukhanov-Sasaki equation, we then read off the scale-factor corresponding to the Hubble-exit of a given mode for the same background.

It is also possible to fix the initial value of A by fixing the present-day scale-factor to be equal to 1, and by fixing the reheating history. Then one can go backward in time to find the corresponding value of the initial scale-factor. However, this is unnecessary, since we are interested in the evolution until the end of inflation, and the inflationary observables are insensitive to A_i once we fix the fact that the pivot scale k_* made its Hubble-exit at a given number of e-folds before the end of inflation (e.g. 60 e-folds in our paper).

4) This is an important point, which is also made by the second referee (Prof. David Wands). Our response is:

We agree with the referee(s). We have added footnote 14, discussing this point. We should also stress that our numerical framework is flexible enough and the user can simulate a mode for much longer duration, rather than the 5-6 e-folds prescription we had mentioned in the paper. We have highlighted this in footnote 14 and mentioned it again on page 21.

Please note that for the specific case of the bump model, 5 e-folds of expansion before and after the Hubble-exit is enough, since it is well-known that the ultra slow-roll phase lasts for less than 3 e-folds (in order to avoid over-production of PBHs)

---

## Round 1 · Referee Report · David Wands (Referee 2) · 2024-6-13

Strengths

1 - clear pedagogical review of primordial power spectra generated by a period of inflation in the very early universe
2 - calculations of both background (homogeneous and isotropic) solutions, and linear perturbations, in minimally-coupled, single-field models of inflation
3 - focusses on the topical subject of ultra-slow-roll models of inflation
4 - some exploration of the efficiency of numerical schemes making different choices for the variables used to track the evolution of perturbations

Weaknesses

1 - there are more sophisticated codes already publicly available, e.g., Py- and M-transport, able to calculate perturbation spectra in multi-field inflation and beyond linear order

Report

Overall this paper represents a useful addition to the literature and with some improvements listed below it should be suitable for publication in SciPost

Requested changes

1- Early on (in the text after eq.12) the authors fix the CMB scale to 60 e-folds before the end of inflation. In fact the CMB scale is dependent on the duration of reheating and this is a major uncertainty in observational predictions in particular models. This deserves more consideration by the authors. At the very least, as a limitation of the present analysis it should be noted. This also applies to points 4 and 5 on pages 15 and 16 (section 4).

2- Near the start of section 3, the authors comment “two gauge-independent massless fields, one scalar, and one transverse and traceless tensor, are guaranteed to exist during inflation”. I found this a bit confusing. There are two massless, transverse and traceless degrees of freedom (the free gravitational field) but the scalar degree of freedom is not strictly massless; it is only approximately massless during slow-roll inflation.

3- There is a factor of a (the scale factor) missing in the last equation in eq.(19).

4- The right-hand-side of Eq.(27) must be evaluated at Hubble exit, k=aH. In general it will evolve on super-Hubble scales, whereas the curvature perturbation remains constant on super-Hubble scales.

5- Eq.(32) presumably corresponds to the specific case of a gravitational wave propagating along the direction (0,0,1). This should be noted. Eq.(33) is a more general form.

6- The normalization of the polarization tensors, εij, in the text following eq.(33) seems unusual. Is the factor 2 correct? It is purely conventional, of course, if followed consistently, but usually one chooses orthonormal polarisation tensors (which would make the factor of the square-root of 2 unnecessary in eq.(33)).

7- The consistency relation (47) is described as a “smoking-gun test of inflation” but it only holds for the specific class of single, canonical field (with sound speed equal to the speed of light), slow-roll inflation, and can be broken in models with multiple fields or non-canonical sound speed. These caveats should be given.

8- There are a few typos that should be corrected, such as the extra “)” in the text after eq.(20) and the extra “In fact, “ between eqs.(50) and (51).

9- After eq.(56), and again at the start of section 3.2, the authors state that observations favour an “asymptotically-flat” potential, which is overstating what observations can tell us. Observations only probe a finite range of the potential, not the asymptotic behaviour.

10- The authors suggest at the start of section 3.2 that the low tensor-to-scalar ratio provides strong evidence for single-field inflation. Why? For example, additional fields tend to suppress the tensor-to-scalar ration (by providing additional scalar sources). So a low-tensor-to-scalar ratio could be considered evidence for multiple fields or a low effective sound speed.

11- Eqs.(75-77) apply only for perturbations evaluated at Hubble-exit during slow-roll inflation, in which case they were previously given in Eqs.(42-45) so they seem out of place in section 4 if the numerical results are not limited to slow roll.

12- In section 5, between Eqs.(82) and (83), the authors say that mode functions for perturbations only need to be evolved for a few e-folds around Hubble exit. But in non-slow-roll inflation models, especially models with a sudden, non-adiabatic transition, perturbation modes can be affected on scales far from the Hubble scale. See for example Jackson et al, arXiv:2311.03281. This also relates to point 5, page 22, where the authors evolve the mode function until 5 e-folds after Hubble exit. This may be long enough for the models considered by the authors, but why not evolve until the end of inflation? Or leave the number of e-folds after Hubble exit as a user-specified parameter.

13- Figure 18, in the caption, the authors claim that the power spectrum at CMB scales matches that of the base KKLT model, also shown. While the power at the CMB scale looks the same, the tilt appears to be different. I would certainly it expect it to be different because of the additional 15 e-folds that occurs due to the presence of the bump, changing the relationship between the background field value and the number of e-folds with respect to the base model (see point 1 above).

14- Note that Ananda et al, astro-ph/0612013, studied the induced GW background from primordial perturbations in the radiation era, after Matarrese et al [58] but shortly before Baumann et al [59].

15- There are a some limitations of the code which the authors should consider:

- The numerical code uses cosmic time as the time variable. This makes it harder to estimate the interval required to simulate the observable duration of inflation. The logarithmic expansion (“e-folds”) is usually more useful for this purpose.

- It would be helpful to provide a user guide to the code with expected example outputs using jupyter notebooks for example.

- There is currently no way for the used to verify outputs.

Recommendation

Ask for minor revision

  • validity: high
  • significance: ok
  • originality: ok
  • clarity: high
  • formatting: excellent
  • grammar: perfect

Author:  Swagat Saurav Mishra  on 2024-09-15  [id 4783]

(in reply to Report 2 by David Wands on 2024-06-13)
Category:
remark
answer to question
correction

Dear Referee, We are thankful for your detailed and insightful comments and suggestions on the earlier version of our manuscript. We have made an honest attempt to take all of your suggestions into account in the revised version of the manuscript. In particular, the following is our response to your comments and suggestions.

1) Regarding the number of e-folds before the end of inflation, corresponding to the Hubble-exit of the CMB pivot scale, we agree and accordingly we have added notes at the respective places in the revised manuscript.

2) We agree, that the phrase involving 'two gauge-independent massless fields' was confusing, therefore, we have edited the text accordingly.

3) Indeed, the scale factor 'a' was missing in Eq. (19), this has been corrected.

4)Thank you for this important comment. We have now explicitly mentioned below Eq. (27) that H and $\epsilon_H$ must be computed at the Hubble-exit time of k.

5) We have mentioned the convention (assumption) used in Eq. (32) in the revised manuscript as a footnote. We have also noted that Eq. (33) is more general. Thanks for this suggestion which brings more clarity to the text of our paper.

6) The factor of 2, and therefore our field redefinitions, are indeed less conventional. However, we rechecked the formulae and calculations and we found that what we had written was correct. Therefore, we have kept it unchanged in the revised manuscript.

7) We agree with the referee's suggestion here. therefore we have changed the text below Eq. (47) and we have also added a footnote highlighting the point made by the referee.

8) We apologize for the typos and thank the referee for pointing them out. They have been corrected in the revised manuscript.

9) Following the referee's suggestion, we have changed the text below Eq. (56), highlighting this point.

10) We agree with the referee's point here, the statement is valid only for canonical single field models. Therefore we have modified the text at the start of sec. 3.2 taking the referee's suggestion into account.

11) We have now made it explicit that the formulae given in Eqs. (75), (76) and (77) are valid under the slow-roll approximations and hence they are applicable only for the potentials discussed in sec. 4.2.

12) We agree with the referee. We have added footnote 14, discussing this point. We should also stress that our numerical framework is flexible enough and the user can simulate a mode for much longer duration, rather than the 5-6 e-folds prescription we had mentioned in the paper. We have highlighted this in footnote 14 and mentioned it again on page 21.

13) This is an important point and we completely agree with the referee. Therefore, we have removed the extra annotation from fig. 18 and highlighted this point in the caption of that figure. We have further referred the reader to Mishra and Sahni 2019 paper (https://arxiv.org/abs/1911.00057) where this point was discussed more explicitly.

14) We apologize for missing out such an important early paper on SIGWs. We have included it in the revised manuscript and we thank the referee for pointing this out.

15) Regarding the limitations of our numerical framework

a) We have added a long paragraph in the discussion section commenting on the usage of cosmic time 't' as the time variable and the potential advantages and disadvantages over the usage of number of e-folds 'N'.

b) We have now included an IPython notebook as a user guide to implement our numerical framework and we have provided a link to the notebook (in the same GitHub account) in the revised manuscript. We thank the referee for this important suggestion.

In response to the weakness of our framework : We completely agree with the referee that there exist more sophisticated numerical codes in the literature. We have added a discussion on this in the discussion section of our revised manuscript and included two new references to PyTransport and CppTransport frameworks.

However, we would like to stress that our framework is very simple and user-friendly. It is quite fast and works very well for computing two-point functions. We are keen on extending the framework to compute higher-point functions to study non-Gaussianity and we will be happy to report on the chanllenges and results from our analysis in the future.

---

## Round 2 · Referee Report · David Wands (Referee 2) · 2024-9-20

Report
Recommendation
Publish (meets expectations and criteria for this Journal)

---

## Round 2 · Referee Report · L. Sriramkumar (Referee 3) · 2024-10-12

Strengths
- As I had said in my original report, the manuscript is written clearly.
- It deals with a topic that is important in the context of primordial cosmology, viz. the dynamics of inflaton and generation of perturbations during inflation.
- The manuscript discusses models/scenarios that have drawn attention in the recent literature.
Weaknesses
- I had hoped that, in the revised version, the authors will use e-folds as the independent variable. But, the authors have continued with cosmic time. I believe this limits the efficiency of the code.
Report
addressed the other points I had raised. They have specifically
commented on my last point concerning the evaluation of the
power spectrum (in non-trivial scenarios) at suitably late
times. So, I recommend the manuscript for publication.
Recommendation
Publish (meets expectations and criteria for this Journal)

---

## Round 2 · Referee Report · L. Sriramkumar (Referee 3) · 2024-10-12

Strengths
- As I had said in my original report, the manuscript is written clearly.
- It deals with a topic that is important in the context of primordial cosmology, viz. the dynamics of inflaton and generation of perturbations during inflation.
- The manuscript discusses models/scenarios that have drawn attention in the recent literature.
Weaknesses
- I had hoped that, in the revised version, the authors will use e-folds as the independent variable. But, the authors have continued with cosmic time. I believe this limits the efficiency of the code.
Report
Recommendation
Publish (meets expectations and criteria for this Journal)

---

## Round 2 · Author Response

Thanks a lot for sending us the reports by the two referees. The comments and suggestions made by the two referees were insightful and detailed. We have made a sincere attempt to address them in the revised manuscript, and all changes appear in violet colour text. The suggestions have greatly improved the quality of our paper, for which we are thankful to the referees . We hope the revised version will be suitable for publication at the Scipost Physics Codebases.
With Regards,
-Swagat

---

## Round 2 · List of Changes

In reference to the comments and suggestions made by the two referees, we have made the following changes which appear in violet colour text in the revised manuscript.
1) On page 4, in the paragraph below Eq. (12), we have added a statement -
'While we have fixed N∗ = 60 for the most part of this work, it is important to note that the exact value of N∗ depends upon the post-inflationary reheating history [16].'
2) At the beginning of sec. 3 on page 5, we have included -
'At linear order in perturbation theory, one gauge-invariant scalar degree of freedom (which is approximately massless during slow-roll inflation), and two gauge-invariant (transverse and traceless) massless tensor degrees of freedom are guaranteed to exist in the single-field inflationary paradigm [53, 54].'
3) On page 6, in the paragraph below Eq. (27), we have included -
'where, H and $\epsilon_H$ appearing in the right-hand side of the above equation should be calculated at the time of Hubble-exit of the mode k, namely, when k = aH. '
4) We have added footnote 2 on page 7, which reads
'Note that in Eq. (32) we have assumed the tensor modes to be propagating along the z-direction, i.e. along (0,0,1); while Eq. (33) is valid in general, independent of the aforementioned assumption.'
5) On page 8, in the paragraph below Eq. (47), we have included
'for the slow-roll inflationary paradigm of a single scalar field with a canonical kinetic term' ;
along with footnote 4 on the same page, which reads -
'Note that the consistency relation does not hold for a non-canonical scalar field for which the speed of sound $c_s^2 ̸= 1$, as well as for multi-filed inflation, in general.'
6) On page 9, in the paragraph below Eq. (56), we have included the phrase -
'...for simple slow-roll potentials that do not possess any features on small scales outside the CMB window,'
7) In the last paragraph on page 9, we have noted -
'Furthermore, the current obser vational constraints [66] are consistent with predominantly Gaussian primordial fluctuations. Within the canonical single-field inflationary paradigm, this provides support... '
8) On page 13, in the paragraph below Eq. (77), we have included the comment -
'where the last three Eqs. are valid only under the slow-roll approximations, as discussed in Sec. (4.2).'
9) We have added footnote 11 on page 15, which reads -
'We again stress that the exact value of $N_e$ depends upon the reheating history in the post-inflationary universe. While we fix it to $N_e = 60$ for the purpose of illustration, in principle, our numerical framework allows for incorporating a different value of $N_e$ without any trouble.'
10) On page 19, in the paragraph below Eq. (82), we have added the phrase -
' for a relatively shorter duration of time';
along with footnote 14 on the same page, which reads -
'Keeping in mind the important caveat that the initial time should be sufficiently early enough to impose Bunch-Davies initial conditions, and the final time should be sufficiently late enough for the mode to be frozen outside the Hubble radius. As discussed below, for most of the potentials considered in this work, as well as for most single-field slow-roll violating models [68] relevant for PBH formation in the literature, imposing initial conditions for about 5-6 e-folds before the end of inflation is sufficient.
Nevetheless, in general, especially for slow-roll violating models, one might need to evolve the mode functions for longer duration and our numerical set-up easily allows the user to incorporate a longer evolution duration for the mode functions.'
11) On page 21, at the end of the paragraph within the point 3., we have included the caveat -
' (however, see the caveat given in footnote 14).'
12) We have removed an annotation from figure 18 and we have added the following to the caption of figure 18-
'the amplitude of the power spectrum roughly matches that of the base KKLT potential, although its spectral index is shifted due to a gain in the number of e-folds because of the presence of the bump, see Ref. [44] for a detailed discussion on this.'
13) In the discussion section, at the end of page 30 and at the beginning of page 31, we have included the following comment -
'...even more sophisticated, frameworks relevant to inflation have been suggested in the literature. For example, Ref. [140] discusses a python package called PyTransport, while Ref. [141] discusses a C++ based platform called CppTransport for the numerical computation of inflationary correlators. Similarly,...'
14) We have added a new paragraph on page 31 in the discussion section, which reads -
'Furthermore, it is important to note that our work uses cosmic time t (along with a suitable mass scale to generate a dimensionless variable) as the time variable. In the literature, most frameworks incorporate number of e-folds N as the time variable since it is arguably more intuitive in describing the evolution during inflation. On the other hand, the usage of cosmic time t is quite convenient under certain circumstances. Firstly, for quasi dS (near-exponential) inflation, since H is almost constant, N ≃ Ht, therefore both N and t are equally intuitive, especially for vanilla slow-roll models. Additionally, cosmic time is particularly more suitable for describing the post-inflationary oscillations, and hence it can also be used conveniently to simulate the dynamics scalar-field dark matter. Moreover, our framework also describes the study of initial conditions for inflation, where the initial phase-space trajectories may correspond to pre-inflationary (decelerating) epoch. Nevertheless, given a vast majority of numerical frameworks on inflation do utilize N , our work provides an alternative time variable. However, it is worth pointing out that computing higher- order correlators as well as quantities at a higher-order in perturbation theory (loop corrections to power spectra), the cosmic time variable might present challenges. Similarly, it may be important to investigate the efficiency of cosmic time as a time variable while comparing models against cosmological data. We plan to return to these issues in the near future.'
15) We have included the references [52,60,140,141] in the revised manuscript.
16) We have created an Python Notebook as a user guide in our GitHub folder (with link https://github.com/bhattsiddharth/NumDynInflation/blob/main/Num_Dyn_Inflation.ipynb)
17) We have added a scale factor 'a' in the final expression of Eq. (19).

---

## Editorial Decision

published